# Biotic interactions promote local adaptation to soil in plants

Thomas Dorey [1,3], Léa Frachon [1,4], Loren H. Rieseberg [2], Julia M. Kreiner [2] & Florian P. Schiestl [1]✉

Although different ecological factors shape adaptative evolution in natural habitats, we know little about how their interactions impact local adaptation. Here we used eight generations of experimental evolution with outcrossing *Brassica rapa* plants as a model system, in eight treatment groups that varied in soil type, herbivory (with/without aphids), and pollination mode (hand- or bumblebee-pollination), to study how biotic interactions affect local adaptation to soil. First, we show that several plant traits evolved in response to biotic interactions in a soil-specific way. Second, using a reciprocal transplant experiment, we demonstrate that significant local adaptation to soil-type evolved in the "number of open flowers", a trait used as a fitness proxy, but only in plants that evolved with herbivory and bee pollination. Whole genome re-sequencing of experimental lines revealed that biotic interactions caused a 10-fold increase in the number of SNPs across the genome with significant allele frequency change, and that alleles with opposite allele frequency change in different soil types (antagonistic pleiotropy) were most common in plants with an evolutionary history of herbivory and bee pollination. Our results demonstrate that the interaction with mutualists and antagonists can facilitate local adaptation to soil type through antagonistic pleiotropy.

Adaptation is a key process in evolution, leading to the emergence and modification of traits and the macroevolutionary diversification of organisms[1]. Adaptation at the population level is typically associated with evolutionary genetic changes that optimize the performance of organisms in their specific habitat, a phenomenon called local adaptation[2,3]. Local adaptation can be due to conditional neutrality, in which alleles are adaptive in one habitat, but neutral in the other, and/ or antagonistic pleiotropy (i.e., a genetic trade-off), where different alleles at a locus are favored in different environments[4–7]. The relative importance of these mechanisms in local adaptation remains unclear[8,9]. Antagonistic pleiotropy is considered critical for maintaining genetic variation among natural populations, but direct experimental proof of this mechanism remains scarce[6–8,10–12]. To detect the genomic basis of local adaptation, most studies use reciprocal transplant experiments and measure fitness-marker associations in

organisms in the local versus a foreign environment[13]. The detection of genetic trade-off is, however, hampered by the fact that fitness-marker associations must be significant in two environments and that temporal variation in environmental conditions may mask such associations during a particular season[7]. Therefore, long-term field studies or experimental evolution approaches provide a more powerful approach to detect antagonistically pleiotropic loci[14]. For example, in an 8-year field study on *Arabidopsis thaliana* conducted in Italy and Sweden, four fitness QTLs displayed a pattern of antagonistic pleiotropy, and seven showed that of conditional neutrality[8]. Other studies have used herbarium genomics or resurrection experiments to detect antagonistic selection in loci across populations, leading to strong population differentiation for certain loci[15,16]. Also, the genomic basis of local adaptation, whether it typically involves few versus many genes, and their relation to phenotypic traits is poorly

[1]Department of Systematic and Evolutionary Botany, University of Zürich, Zürich, Switzerland. [2]Department of Botany and Biodiversity Research Centre, University of British Columbia, Vancouver, Canada. [3]Present address: Department of Environmental Sciences, University of Basel, Basel, Switzerland. [4]Present address: Agroécologie, INRAE, Institut Agro, Univ. Bourgogne, Univ. Bourgogne Franche-Comté, Dijon, France. ✉e-mail: florian.schiestl@systbot.uzh.ch

understood[17]. Besides maintaining genetic diversity, local adaptation may also promote ecological speciation, through the emergence of ecotypes and eventually isolating barriers between differently adapted populations[18,19].

Local adaptation in plants occurs through shifts in phenotypic and phenological traits and has been detected in response to climate variation[16,20,21] or soil, driven by chemical factors such as salinity[22], carbonate[23], or heavy metal content in serpentine soils[24]. The role of biotic factors in local adaptation, such as mycorrhizal mutualists[25], herbivory[26] or pollination[27] is much less well understood[28,29]. Plants can locally adapt to biotic interactors, if such interactions are spatially heterogeneously distributed across the landscape, in a geographic mosaic-like fashion. In plants, such local adaptation to the pollinator- or herbivore "climate" (sensu[30]) has been shown previously and is thought to be the basis for population variation in floral traits and defense mechanisms[27,31]. Besides driving local adaptation directly, biotic factors may also interact with abiotic factors and thus modify the selection strength and adaptive response of plant populations in an indirect way[29]. For example, the availability of soil nutrients may modify trade-offs between defense against herbivores and the ability of a plant to re-grow any lost tissue (i.e., tolerance). For this reason, herbivory may primarily select for resistance in nutrient-poor soils, whereas in nutrient-rich soils, it may favor tolerance (i.e., re-growth of lost tissue). This concept is called the growth-defense trade-off[32,33]. Despite the importance of multiple ecological factors for local adaptation, few studies consider more than one factor and biotic interactions such as herbivory or animal pollination are often not included[29,34]. Hence, we lack even a basic understanding of how multiple ecological factors interact to shape local adaptation[35]. In this study, we tested the hypothesis that biotic interactions, by changing patterns of selection and evolutionary trade-offs, impact the local adaptation of plants to soil type.

Soil is a key ecological factor for plants, as it forms a main source of nutrients and water, and enables interactions between plants and soil microbiomes[36]. Many studies have shown that plants adapt to different soil types, through physiological and morphological mechanisms, leading to the formation of soil ecotypes[37]. Sometimes, these soil ecotypes co-vary with floral- and defense traits suggesting that adaptation to soil types can be linked to adaptation to biotic interactions (i.e., pollinators and herbivores)[32,38]. We thus focused our investigation on local adaptation to soil type and the indirect effects of biotic interactions to this kind of adaptation. We focused on the following specific questions: 1) How fast does local adaptation evolve? 2) Is local adaptation to soil type impacted by selection imposed through herbivory and/or bee-pollination? 3) Is local adaptation in our system primarily driven by antagonistic pleiotropy or conditional neutrality at the genomic level?

We used experimental evolution with fast cycling *Brassica rapa* plants and a 3-way factorial design with two different soil types (limestone- and tuff soil, collected in nature and not sterilized), with- and without aphid herbivory, and with either bumblebee- or hand-pollination (i.e., eight treatment groups, Supplementary Fig. 1). Each treatment was replicated two times with 49 plants in each replicate evolving independently during eight consecutive generations, leading to a total number of 784 plants per generation. Eight generations with selection were followed by two generations without insects (i.e., hand-pollination) to reduce maternal effects caused especially by aphid-herbivores. First, to assess phenotypic evolution, plants of generation one and ten of all treatment groups were grown in the soil type they had evolved in, in the greenhouse. Plants were only grown in their "local" soil for this analysis to avoid any soil-induced plastic effects and thus allow for a (within soil-type) comparison of trait evolution. We quantified several plant traits to gain an as-comprehensive-as-possible view of the phenotypic changes that may have evolved.

Second, we conducted a reciprocal transplant experiment, where plants of generation ten of all treatment groups were grown in the soil they had evolved in (local soil), as well as the other soil type (foreign soil). To assess evolutionary changes, plants of generation one were re-grown on both soils along with plants from the tenth generation (i.e., as a resurrection experiment). Altogether in this experiment, 1376 plants were grown and phenotyped, of which 1118 randomly chosen individuals were subsequently genotyped by whole-genome resequencing. In addition, bioassays were performed with bumblebees in flight cages to assess the attractiveness of 912 randomly selected plants using choice assays. All phenotyping and bioassays were done without aphids being present on the plants. To assess local adaptation on the phenotypic level, we chose the trait "the number of open flowers" at pollination day, which had the strongest and most consistent positive association with "relative seed set" and "bumblebee first choices", as assessed in data from generation one and two in this experiment (Supplementary Tables 1, 2). Seed set could not be assessed directly in our local adaptation experiment, because the DNA sampling after the bioassays left many of the smaller plants with a significant proportion of their biomass removed. For the "local" vs "foreign" criteria for local adaptation to be true[3], we expected that plants that evolved in either soil type would always outperform plants that were transplanted into this soil type. Statistically, this would be shown by a significant soil line x soil ( = G x E) interaction and a significant post hoc linear contrast between the "local" (e.g., limestone lines in limestone soil) and "foreign" (e.g., tuff lines in limestone soil) treatments. "Soil lines" were the plant genotypes having evolved in a particular soil, and "soil" was the soil type that plants were grown in during the transplant experiment (i.e., the environment).

In a previous study on phenotypic selection, using plants of generations one and two of this experiment[39], we showed that pollinator-mediated selection varies according to soil type and the presence/absence of herbivory, despite only one type of pollinator being used in this experiment. Selection diverged the most between plants grown in different soils, with herbivory and bee pollination. Based on these different figures of selection, we hypothesized that the strongest local adaptation would evolve in treatment groups with biotic interactions where the stronger divergent selection was observed. Because more fertile soil allows for a more rapid evolutionary response[40], we also expected stronger patterns of local adaptation in tuff soil.

Here we show that local adaptation to different soil types evolved only in plants that interacted with herbivores and bee pollinators. Whole genome re-sequencing of experimental plants revealed that biotic interactions led to a tenfold increase in SNP markers with significant allele frequency change and that plants that had interacted with herbivores and bee-pollinators showed the most markers with opposite allele frequency change in the different soil types (antagonistic pleiotropy). We conclude that biotic interactions speed up the evolution of local adaptation to soil types in plants and that antagonistic pleiotropy is a key mechanism for driving this evolutionary process.

## Results

Plants in every treatment group showed evolutionary changes in many traits, with both increases and decreases being apparent, suggesting patterns of resource reallocation (Fig. 1; Supplementary Tables 3 and 4 and Supplementary Data 1). For example, leaf size decreased in plants in both soil types with an evolutionary history of herbivory but increased in plants in tuff soil without past herbivory and bee pollination (Supplementary Tables 3 and 4). Plants flowered earlier when they had evolved with herbivory in both soil types but flowered later when evolved with bee pollination in tuff soil. The number of open flowers at pollination day increased in plants with past herbivory in limestone soil, and with bee-pollination for plants in tuff soil, and flower size increased with bee-pollination (Fig. 1 and Supplementary

Data 1). Plants with an evolutionary history of bee pollination were more attractive to bees than those with a history of hand-pollination, showing that bumblebee-driven evolutionary changes were adaptive (Fig. 1). For many traits, soil also had a significant effect (Fig. 1, Supplementary Table 3 and Supplementary Data 1), but because for this analysis only plants growing in their local soil were used, these effects could have been caused by either soil-induced plasticity (Dorey and Schiestl 2022) or evolutionary changes.

In the reciprocal transplant experiment, we found a pattern of local adaptation in the "number of open flowers" strongly statistically supported, but only in the plants that evolved with bee pollination and herbivory (Fig. 2D, Table 1 and Supplementary Data 1). In this plant group, attractiveness to bumblebees in plants grown in tuff soil was much higher in local (i.e., tuff-lines) than foreign (i.e., limestone-lines) plants (Supplementary Fig. 2, Supplementary Data 1). Besides the number of open flowers, a few other traits (time to first flower, petal length) also showed a significant soil line x soil interaction in the bee pollination with the herbivory treatment group, and for "time to first flower", the pattern matched the local vs. foreign criterion (Supplementary Table 6 and Supplementary Data 1). Plant height also showed a similar pattern, but the interaction soil lines x soil was only marginally significant (Supplementary Table 6 and Supplementary Data 1). Because we do not have data showing that patterns of evolution are adaptive for those traits, these patterns are, however, not conclusive proof of local adaptation.

For the analyses of local adaptation at the genomic level, we assessed patterns of allele frequency (AF) change in single nucleotide polymorphisms (SNPs) between generation one and ten. To match genomic and phenotypic data, we calculated breeding values for "number of open flowers" and focused our investigation only on markers with breeding values for "number of open flowers" and the associated candidate genes. We found that plants in treatment groups with biotic interactions, especially those with bee pollination, had an order of magnitude more SNPs with significant AF changes (i.e., unlikely due to random change or genetic drift based on False Discovery Rate (FDR) corrected Cochran–Mantel–Haenszel (CMH) tests; see methods), than the one lacking biotic interaction. Plants with herbivory and hand-pollination (HH) had 672, plants without herbivory and bee-pollination (NHB) had 1376, and plants with herbivory and bee-pollination (HB) had 861 significant SNPs. Plants without biotic interactions (no-herbivory and hand-pollination, NHH) had merely 103 significant SNPs (Fig. 2). Only plants with bee-pollination (HB and NHB) had SNP markers that showed a pattern of antagonistic pleiotropy (AP, i.e., opposite direction of AF change in plants evolved on different soil types). Consistent with patterns of local adaptation in the phenotypic data, plants with an evolutionary history of bee pollination and herbivory (HB) showed a trend towards more SNPs with an AP pattern than plants with an evolutionary history of bee pollination without herbivory (NHB) (HB: 24 SNPs, 2.8% of all significant markers; NHB: 20 SNPs, 1.5% of all significant markers, $Chi^2_1 = 7.48$, $P = 0.087$). The effect sizes of those markers also reinforced their importance for local adaption in the HB treatment. For this treatment group, 12 markers had positive- and 12 negative effect sizes for limestone lines, and 16 positive- and 8 negative effect sizes for tuff lines. For the NHB treatment group, 6 had positive- and 14 negative effect sizes for limestone lines, and 2 positive- and 18 negative effect sizes for tuff lines. Mean effect sizes for AP markers in HB were positive in both soil lines and in NHB negative in both soil lines. There was, however, no significant difference in mean effect sizes for AP-markers between the HB and NHB treatments ($F_1 = 2.62$, $P = 0.113$ lime, $F_1 = 0.29$, $P = 0.594$ tuff). In the treatment group HB, AP-markers spanned a larger number of genes (18 genes) than in the treatment group NHB (14 genes; Fig. 2 and Supplementary Table 10). Interestingly there was no overlap in the genes associated to AP markers in the two treatment groups. However, there was a significant overlap between replicates of the two treatment groups, as only markers that showed an AP pattern in both replicates were considered in the analysis. In HB, the putative functions of "AP-genes" include flowering time regulation and response to low nutrient- or otherwise stressful soil types (Supplementary Table 10). For the NHB group, fewer genes had a defined function, including osmotic stress tolerance and photosynthesis (Supplementary Table 10). For conditional neutrality (CN, i.e., significant AF change in one soil type, but no change in the other), the pattern of markers was different. The highest percentage of CN markers was found in the treatment group without biotic interaction (NHH; 41, 41% of significant markers), followed by HH (233, 34%), HB (165, 19%), and NHB (247, 18%). The percentage of markers with positive/negative effect sizes were as follows: (NHH: 44/56 lime, 56/44 tuff; HH: 80/20 lime, 13/87 tuff; HB: 73/23 lime, 64/36 tuff: NHB: 32/68 lime, 33/67 tuff). Mean effect sizes were: NHH: negative for lime and positive for tuff, HH: positive for lime,

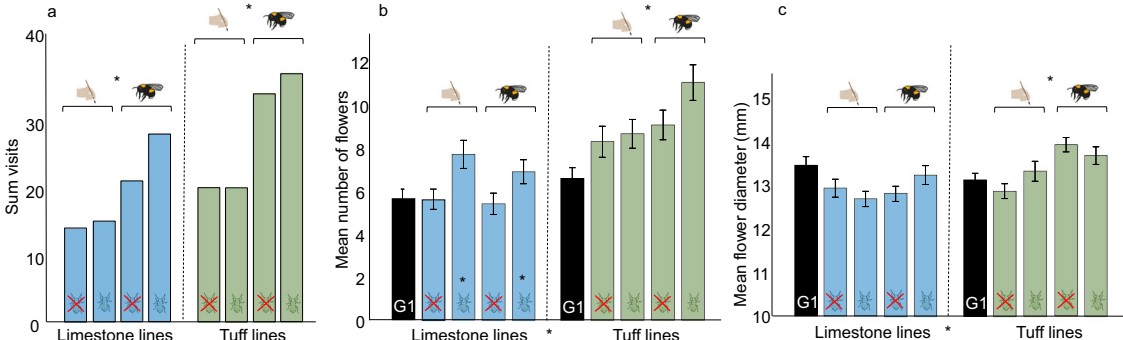

**Fig. 1 | Trait evolution in the evolution experiment.** The figure shows the difference in phenotype (mean ± s.e.m. values) among treatments in plants of generation ten when growing in "local" soil (gray bars), and generation one (black bars; plants of generation one and ten were grown together). Differences in phenotype were estimated by two-sided linear mixed models (LMM). To discriminate evolutionary changes from changes due to soil-induced plasticity, we considered only plants grown in their local soil in this analysis. **a** Plants' attractiveness to bumblebees, (**b**) number of open flowers on the day of pollination, and (**c**) flower diameter. Plants with an evolutionary history of bumblebee pollination were more attractive to bumblebees (**a**) consistent with earlier studies[47,57], had more open flowers (**b**) and produced larger flowers (**c**) in tuff soil than plants that evolved with hand-pollination. Plants in limestone soil that evolved with herbivory had more open flowers than those without herbivory (**b**). Asterisks indicate statistically significant effects of soil, herbivory, and pollination ($P < 0.05$; further statistical values are shown in Supplementary Tables 3–4). The dashed lines distinguish plants growing in limestone (left side) from those growing in tuff (right side). Attractiveness of plants was tested separately for soil types and herbivory groups, because of the strong differences in attractiveness between them (plants in tuff soil and those that evolved without herbivory were generally the most attractive). Attractiveness ($N = 189$ visits for 456 plants) and trait values ($N = 443$ limestones, 456 tuff) were assessed without aphids on the plants (i.e., without the effects of herbivore-induced plasticity).

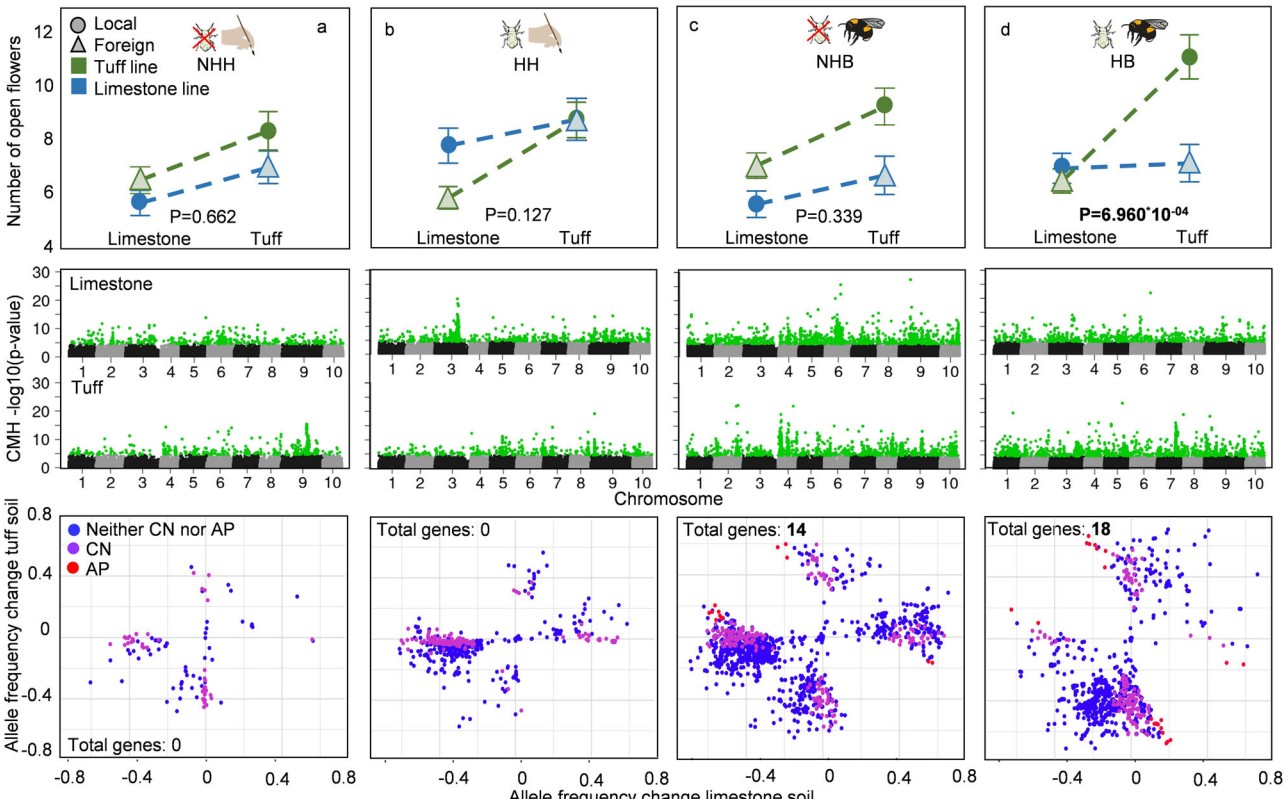

**Fig. 2 | Patterns of local adaptation at the phenotypic and the genomic level across eight experimental treatment groups that varied in their abiotic and biotic factors (two soil types per pollination/herbivory group). First row:** Local adaptation in "number of open flowers" ( ± s.e.m. values); the number of open flowers is shown for plants that evolved in the two soil types (limestone-line, tuff-line) when grown in either their "local" (circles/filled) or "foreign" (triangles/hatched) soil type (the soil used for growing is indicated on the x-axis). Differences in flower numbers between local and foreign lines were estimated by two-sided linear mixed models (LMM). *P*-values give the significance of the soil lines x soil ( = G x E) interaction; significant local adaptation (i.e., significant G x E interaction and local vs foreign contrast; see also Table 1) was only detected in the treatment with an evolutionary history of both herbivory and bee-pollination (panel d). **a** plants that evolved without herbivory and with hand-pollination (*N* = 313). **b** plants that evolved with herbivory and hand-pollination (*N* = 305). **c** plants that evolved without herbivory and with bee pollination (*N* = 296). **d** plants that evolved with herbivory and bee pollination (*N* = 298). For all values and statistics, see Table 1 and

Supplementary Tables 6–9). **NHH:** plants that evolved without herbivory (H) and with hand-pollination (H). **HH:** plants that evolved with herbivory (H) and with hand-pollination (H). **NHB:** plants that evolved without herbivory (NH) and bee pollination (B). **HB:** plants that evolved with herbivory (H) and bee pollination (B). **Second row:** Signals of genome-wide adaptation; plots of CMH p-values of all SNPs across ten chromosomes, separately for plants evolved in different treatment groups. SNPs highlighted in green are those with FDR < 0.05, depicting significant non-random allele frequency change. Third row: Patterns of allele frequency change between generation one and ten in plants in the different treatment groups. Only markers with breeding values for "number of open flowers" (i.e., an association to this trait), and significant allele frequency change in both replicates were included in this analysis; allele frequency change is shown as mean values between the replicates. Red points are markers that fit the pattern of antagonistic pleiotropy (AP) in both replicates, purple are those that correspond to conditional neutrality (CN) in both replicates and blue those that fit neither the criteria for AP nor CN. "Total genes" refers to the number of genes associated to antagonistic pleiotropy.

negative for tuff, NHB: negative for both soil lines, HB: positive for both soil lines. Because these values match the pattern of local adaptation in the different treatment groups found in the phenotypic data less well (i.e., significant local adaptation only in HB), we assumed that local adaptation was driven primarily by AP rather than CN.

## Discussion

Local adaptation is mediated by many features of the environment, because selection in natural habitats is usually driven by a set of interacting factors, including abiotic and biotic ones[28,41]. Nevertheless, the traditional research focus in local adaptation has been on abiotic factors such as climate, or mixed abiotic-biotic factors such as soil[29], even though biotic factors such as herbivores, pollinators, microbiota, or competitors are equally or sometimes even more important than abiotic factors for plant survival and reproduction[28,42–45]. How quickly organisms adapt to the multivariate local habitat conditions[15,35], and which genomic changes enable such adaptation[17] is not well known, despite the importance of rapid adaptation to the survival of many organisms in the face of regional- and global change. Evidence from

locally adapted plants to recently contaminated soils[46] and experimental evolution/resurrection studies examining adaptation to various environmental factors[16,47,48], however, suggest that adaptation can be a rapid process in plants. Whether the speed of adaptation increases or decreases with the number of selective factors likely depends on whether those factors act synergistically or antagonistically on traits, but this is rarely characterized in studies of local adaptation[35].

Because soil is a key factor for plant growth, it is also expected to have a major impact on plant evolution, however, modified by biotic interactions. A recent study analyzing data of plants grown in a common garden, originating from the same evolution experiment as the ones analyzed here, has indeed shown that divergent evolution of traits in response to different soil types is particularly strong with bee- rather than hand pollination[40]. Our phenotypic evolution data shows significant interactions between "soil lines" (i.e., the soil plants evolved in) and either herbivory or pollination for many traits for plants grown in their local soil (i.e., without local transplant data; Supplementary Tables 3, 4), indicating that soil had considerable overall effects on

**Table 1 | Statistical analyses of local adaptation in the trait "number of open flowers" at pollination day in plants in the different treatment groups**

| Number of open flowers | | | | | | | | | |
|---|---|---|---|---|---|---|---|---|---|
| | **Herbivory** | | | | | **No Herbivory** | | | |
| | **Bee-pollination (HB, N = 298)** | | | **Hand-pollination (HH, N = 296)** | | **Bee-pollination (NHB, N = 305)** | | **Hand-Pollination (NHH, N = 313)** | |
| Parameter | df | chi$^2$ | P | chi$^2$ | P | chi$^2$ | P | chi$^2$ | P |
| Soil lines (SL) | 1 | **14.17** | **1.667\*10$^{-04}$** | 0.08 | 0.781 | **7.87** | **0.005** | **7.86** | **0.005** |
| Soil (S) | 1 | **7.06** | **0.008** | **4.59** | **0.032** | **11.90** | **5.628\*10$^{-04}$** | 3.67 | 0.055 |
| Replicate | 1 | 0.06 | 0.803 | 0.01 | 0.937 | 0.78 | 0.376 | 1.60 | 0.206 |
| (SL) x (S) | 1 | **11.50** | **6.960\*10$^{-04}$** | 0.28 | 0.594 | 0.92 | 0.338 | 0.19 | 0.662 |
| *Post hoc tests* | | | | | | | | | |
| | **Parameters** | | **Treatments** | | ***n*** | ***t* – value** | | **P** | |
| Number of open flowers | «foreign» versus «local» | | HB | | 298 | -3.39 | | 7.916\*10$^{-04}$ | |
| | «home» versus «away» | | HB | | 298 | 2.68 | | 0.008 | |

The table gives the values and the statistical parameters of a general linear model with the number of open flowers as dependent variable, soil lines, soil and their interaction as fixed- and replicate as random factor, separately for the herbivory- and the pollination lines (the phenotypic values and detailed statistical parameters for each treatment group are reported in Supplementary Tables 6–9). Post hoc tests (see text for details) were done only for the group with significant soil lines x soil interaction (herbivory & bee pollination). **HB:** plants that evolved with herbivory (H) and bee pollination (B).

how herbivory or pollination determined trait evolution (and vice versa). For example, for plants in limestone soil, the "number of open flowers" increased with herbivory, whereas for plants in tuff soil, this trait increased with bumblebee pollination. This pattern of adaptive evolution mirrors the earlier-documented figures of selection for plants of generations one and two of this experiment[39], where selection on number of open flowers was stronger with herbivory in limestone-soil growing plants, whereas selection on this trait was stronger without herbivory in plants growing in tuff soil[39].

As for the genomic basis of local adaptation to the different soil types, we demonstrate AF changes in many SNPs across the genome. Most strikingly, we show a dramatic increase in the number of loci with significant AF change in treatments with biotic interactions, indicating massive, genome-wide adaptive responses to herbivory and especially to bee pollination. We are only beginning to understand the genomic bases of pollinator- and herbivore-driven adaptation in plants[6,49–51]. A recent genomic study based on another evolution experiment demonstrating rapid adaptation to pollinators in *Brassica rapa*, showed similar polygenic patterns of adaptation to bumblebees[52]. Our results reinforce the finding that biotic interactions cause genome-wide polygenic adaptation in this plant species.

Our reciprocal transplant experiment showed that the interaction of herbivory and bee pollination was also of key importance in driving local adaptation to soil type, and our genomic data suggests that this local adaptation is facilitated by genetic trade-offs. Whether such antagonistic pleiotropy is more commonly the basis for local adaptation than conditional neutrality is still an open question[9]; however, genetic trade-offs in plants are increasingly recognized to play a key role in adaptation to different soil and climate conditions, via flowering, defense, or growth[6,7,10,11,50].

Our results suggest that biotic interactions reinforce trade-offs caused by adaptation to different soil types, thus speeding up local adaptation. Although this has never been experimentally examined before, a previous meta-analysis has shown that in plants, local adaptation to abiotic factors is usually stronger in the presence of biotic factors[28]. This is a plausible scenario, as plants may only be able to respond to pollinator- or herbivore-mediated selection by simultaneously optimizing adaptation to soil-type, and thus nutrient gain and/or soil-specific resource reallocation. Besides optimizing trade-offs between different traits, plants are also selected to increase the resources available to them (i.e., their resource status)[53], and this can be achieved by physiological traits for optimizing nutrient gain. Such traits likely differ between soil types and associated microbes, for

example, calcareous soils such as the limestone soil used here are challenging for plants to grow in because their high concentration of calcium carbonate limits the uptake of essential nutrients such as iron. Thus, optimizing growth in this type of soil needs calcium exclusion and improved iron uptake[23], traits that in other soil types such as tuff soil may unnecessarily slow down growth and thus prevent responses to pollinator- and herbivore selection.

A question that remains open from our experiment is whether soil differences without biotic interactions are sufficient to foster local adaptation, given long enough time spans. Hence, we cannot infer whether biotic interactions merely speed up the local adaptation process or trigger an evolutionary trajectory that is qualitatively different from local adaptation driven by soil differences alone. Field studies will hardly be informative for this question, too, as studies that showed local adaptation or speciation in response to soil type usually do not control for the effects of biotic interactions; such interactions are ubiquitous even for wind- and self-pollinated plants that do not need animal pollinators but will nevertheless interact with herbivores or have done so in their recent past. Thus, whether and when local adaptation evolves without biotic interactions, and how it differs from adaptation in their presence could best be answered with longer-term experimental evolution in the greenhouse.

Our experimental evolution study shows that selection imposed by the pollinator x herbivore interaction can promote local adaptation to the soil through antagonistic pleiotropy. Because local adaptation leads to the formation of ecotypes, it may subsequently cause speciation via pleiotropic effects on reproductive isolation[46,54]. Indeed, ecotypes and species of plants sometimes differ in traits related to soil adaptation as well as the attraction of different pollinators[38], and differences in herbivory are sometimes linked to plant ecotypic differentiation, too[55,56]. Thus, our finding of local adaptation to soil being promoted by biotic interactions suggests that the interplay between biotic and abiotic factors can drive plant diversification. Both biotic pollination and herbivory have been known as drivers of plant adaptation, but their interactive effects on local adaptation and potential speciation provide new avenues into understanding how biotic interactions shape plant diversification.

## Methods
### Study system
We used annual "fast cycling" *Brassica rapa* plants (Fast Plants® Standard Seed with maximum genetic diversity) in this study. These plants are self-incompatible, have a short generation time of about two

months, harbor high genetic variation, and have previously been shown to be suitable for selection- and experimental evolution studies[47,57–59].

## Experimental design

In 2018, we grew 440 seeds of fast-cycling *Brassica rapa* plants obtained from Carolina Biological Supply (Burlington, NC, USA) in a phytotron under standardized soil, humidity, temperature, and watering conditions. Of the 440 seeds, 410 germinated and were used to produce full sib seed families by manually crossing 205 randomly assigned plant pairs. These crossings resulted in a total of 163 seed families (only pairs where both parents produced seeds were considered a seed family in our experiment). Of these 163 seed families, we randomly assigned 98 families to two replicates (A and B) and 8 treatment groups (see Supplementary Fig. 1), so that each replicate included 49 plants (to allow a squared 7 × 7 setup for pollination). In an experiment conducted in 2020, we showed that after eight generations of selection, plants crossed within replicates produced the same amount of seeds as plants crossed between replicates[40]). Plants grown from the seeds of the crosses within and between replicates showed similar phenotypes suggesting little to no inbreeding effect on plant phenotype[40]. Therefore, we assumed that our sample size of 49 per replicate was large enough to avoid any inbreeding effects in the phenotype after eight generations of evolution, and thus no crossings between replicates were deemed necessary for the phenotyping of plants at the end of the experiment. Seed families were equally spread across the whole experiment so that every family was represented in each treatment in one replicate (families 1–49: replicate A, 50–98: replicate B; Supplementary Fig. 1).

Our evolution experiment lasted for eight generations and encompassed three factors: growing in limestone soil or tuff soil, hand-pollination or bee pollination, and aphid herbivory or no herbivory (Supplementary Fig. 1); these factors were combined in a full factorial way leading to eight treatment groups. Each treatment was replicated twice (replicates A and B), and each replicate was kept as an isolated line through the whole experiment to be able to assess independent and reproducible evolutionary changes. During the whole experiment, we first grew the plants in a phytotron under 24 h of light, 21 °C, 60% humidity and watered them once a day (at 8:00 h). Plants grown in limestone soil were sown out four days earlier than plants in tuff soil, because of their delayed development. After repotting, we moved the plants to an air-conditioned greenhouse with natural and additional artificial illumination to achieve growing conditions with 16 h of light and a constant temperature of 23 °C, with uniform watering. Plants were grown in individual pots (7 x 7 x 8 cm), and were fully randomized multiple times during the experiment, except during herbivory treatments. During herbivory treatments, we randomized the placement of plants within each herbivory treatment to minimize any potential cross-contamination effects.

## Experimental evolution and treatments

Our study was designed to examine the effects of soil, herbivory, and bee pollination on plant adaptation, using *Brassica rapa* as a model system. To achieve as-natural-as-possible factors in the experiment, we set up the experiment to mimic natural conditions found in populations of a Southern Italian *Brassica* species, *Brassica incana*. Nevertheless, the objective of the study was not to investigate evolutionary patterns specific to *Brassica rapa* or any other plant species but to provide general insights into how biotic factors shape plant local adaptation, and hence create new testable hypotheses of how different ecological factors interact to drive plant evolution. We used two different soil types for our experiment, either limestone (L) or tuff soil (T), that are found in Campania (Italy) and represent common soil types on which *Brassica incana* grows. We collected soil in two natural populations of *B. incana*, in July and October 2018 at Valico di Chiunze

(40.719 °N, 14.619 °E) for limestone, and at Monte di Procida (40.809 °N, 14.045 °E) for tuff soil. Both soils were gathered from the surface layer (ca. 500 kg per soil type), sifted (using a mesh of 1 cm of diameter), stored in bags and shipped to Switzerland. The soil was not sterilized before use. Because fast-cycling *B. rapa* plants were growing rather poorly in the limestone soil, and plants of generations three to four produced hardly enough seeds per plant to sustain the next generation. Thus, starting from generation five, we added fertilizer to both soil types in the form of 10 ml of an universal garden fertilizer diluted in 10 L of water used for watering (NPK: 8-6-6 with traces of B, Cu, Fe, Mn, Mo, Zn; Wuxal, Maag Agro, Dielsdorf, Switzerland). Nutrient addition during a generation was done once during sowing out and a second time after pricking. Despite our use of fertilizer, we showed that seed production as well as plant growth were still resource-limited in our experiment[40], as plants growing in standardized soil (with optimal fertilization) had higher seed production and higher trait values compared to plants growing in either limestone or tuff soil.

## Herbivory

In addition to soil differences, plants were either exposed to aphid herbivory or kept without herbivory (H: herbivory, NH: no herbivory). Aphids are widespread sucking herbivores, feeding on a great range of plants from gymnosperms to angiosperms[60], and *Brassicaceae*'s specialists such as *Brevicoryne brassicae* are common herbivores on crucifers; in wild *Brassica incana* plants, *B. brassicae* is indeed the most common herbivore[61]. Therefore, we used *B. brassicae* as herbivore in our experiment. We collected this species of aphid in the summer 2018 at the Botanical Garden of the University of Zurich (Switzerland) and then reared them in climatic chambers on fast-cycling *Brassica rapa* plants under 16 h light at a temperature of 23 °C and humidity of 70%. Plant herbivory in our experiment was started at the "two true leaves stage", around 13, 14 days after sowing out for plants growing on tuff, and 17, 18 days after sowing out for plants growing on limestone. Each single plant was infested by adding 10 wingless aphids on the leaves, which were allowed to feed on the plants for 72 h. Previous studies showed that ten aphids induce plastic responses within the hour of the infestation, and different growth rates after 72 h of infestation[61]. During the time of the herbivory, infested and non-infested plants were covered by a net (7x7x15 cm) with mesh size of 680 μm (Bugdorm, model DC0901-W). After the three days of infestation, nets and aphids were carefully removed from the plants and plants were subsequently checked every day for left-over aphids, which were removed as well.

## Pollination

For pollination, we used either hand pollination or pollination by bumblebees (*Bombus terrestris*[57]). *Bombus terrestris* is a common and efficient pollinator of various Brassica species[61]. Bee pollination was done 10 days after herbivory (when all aphids had been removed) and most of the plants were flowering. For each replicate in each generation, pollination was done between 8.30 am and 5.30 pm. Plants were randomly set up in a square of 7×7 in a flight cage (l x w x h: 2.5 m x 1.8 m x 1.2 m) with a 20 cm distance between plants. A total of seven bumblebee workers were used for pollination per replicate. Bumblebees were released individually, and each bee was recaptured after visiting a total of 5 plants; each bee was only used once. The number of visits was limited to ensure pollen limitation of reproductive success at the replicate level. This limitation was necessary because fewer bees will interact with larger natural plant populations than in our experimental conditions, yet in our experiment, the use of several bee individuals was necessary to control for genetic variation among bees. As a consequence of the limitations of visits, in each replicate, around 17 plants produced seeds (replicate A, mean: 17.41 ± 2.63 and range of 11–23 plants), replicate B, mean 16.57 ± 2.27 and range of 12–22 plants). The reason for fewer plants producing seeds than possible with seven

bees and 5 visits by each, is that some plants got multiple visits because they were more attractive than others. This "over-visitation" is relevant for fitness and thus impacts selection imposed by bees. The level of pollen limitation was considered to mirror natural conditions for many plant species; overall, reproduction in outcrossing angiosperm species is commonly limited by pollen receipt[62,63]. For hand-pollination, to achieve the same proportion of plants potentially bearing fruits as in the bee-pollination treatments, we randomly selected 28 mother plants per replicate for pollination. In these control groups, we randomly assigned, one father plant to a mother plant of the same replicate, and of four flowers, one long stamen was sampled and used to deposit an excess of pollen on one stigma on four flowers per mother plant. As the hand-pollination treatment serves as a control for pollinator-mediated selection, the number of flowers pollinated was kept constant to avoid any selection via pollination. As in the bee-pollinated treatments, visited plants had an average visitation between 1 and 2 (mean ±s.d.: TNHB (plants that evolved in tuff with no-herbivory and bee-pollination): 1.55 ± 0.13; THB (plants that evolved in tuff with herbivory and bee-pollination): 1.78 ± 0.22; LNHB (plants that evolved in limestone with no-herbivory and bee-pollination): 1.66 ± 0.23; LHB (plants that evolved in limestone with herbivory and bee-pollination): 1.71 ± 0.25), thus, a single father plant for the hand-pollinated treatments was justified and led to only slightly higher average number of paternity in the seeds of plants in the bee-pollinated treatments. Effective population sizes were calculated as the average of plants producing seed throughout the generations; values were rounded up, as some visitations/crossings do not lead to seed set (because of genetic incompatibilities), despite pollen is exported. Values for Ne used in the CHM tests (see genomics analyses below) were: LNHH (plants that evolved in limestone with no-herbivory and hand-pollination): 28, LHH (plants that evolved in limestone with herbivory and hand-pollination): 26, LNHB (plants that evolved in limestone with no-herbivory and bee-pollination): 22, LHB (plants that evolved in limestone with herbivory and bee-pollination: 22, TNHH (plants that evolved in tuff with no-herbivory and hand-pollination): 28, THH (plants that evolved in tuff with herbivory and hand-pollination): 27, TNHB (plants that evolved in tuff with no-herbivory and bee-pollination): 23, THB (plants that evolved in tuff with herbivory and bee-pollination): 21.

After pollination, we marked the region of the plant with fully open flowers of visited plants. Because flowers open from bottom to top, we marked the lowest and highest open flowers of the inflorescences, and only fruits that developed from between these marks were collected and counted after ripening. Plants were kept in the greenhouse under standardized light and watering conditions for completing fruit development. Four weeks after pollination, plants were deprived of water and started to dry for seed maturation. Once fully dried, fruits and seeds were harvested from the different pollination treatments, counted, and weighted. To grow the next generation, we calculated the seed contribution of each plant to the next generation, to ensure that all individuals contributed proportionally to their total seed production in the replicate to the next generation. The seed contribution of all individuals was adjusted to achieve again a sample size of 49 plants per replicate according to the formula:

$$\text{Plant contribution} = \frac{\textit{Individual seed set}}{\textit{Replicate sum of seeds}} * 49 \quad (1)$$

### Reciprocal transplant experiment
To study whether plants evolved local adaptation to the soil they grew in during our experiment, we performed a reciprocal transplant experiment at the end of the evolution experiment. To reduce maternal effects in seed quality, we grew plants of all treatment groups without herbivory and with hand-pollination for two generations (i.e., generations nine and ten). These plants were grown in their native soil in Generation 9, and seeds produced from hand-pollination in Generation nine were used for the reciprocal transplant experiment and phenotyping of plants done in Generation 10. In the hand-pollination process of generation 9, each plant was used as both a father and a mother plant, albeit in different pairs. The pairing involved the random selection of a father and mother plant within the same replicate.

For the reciprocal transplant experiment, we randomly selected 40 of the 49 seed families per replicate of each treatment group of generation ten. The selected seed families were grown on both limestone and tuff soil. This allowed for a comparison of plant performance in the soil they evolved in and the one they did not, while controlling for genotype (i.e., plant family). To assess the evolutionary changes of plants in generation ten, we grew in parallel two individuals (one on limestone, one on tuff) from 36 families randomly chosen out of the 49 full-sib seed family per replicate used as the starting population (generation one).

### Pollinator preference assays
To assess the attractiveness of plants to pollinators in native and non-native soil, we performed pollinator preference assays. Attractiveness to pollinators, assessed as the first choice of bees, was used as a fitness proxy in our study. *Brassica rapa* is an outcrossing species and therefore its reproductive success is highly correlated to pollinator visitation (correlation between the number of visits and the number of seeds: $r_{784} = 0.62$, $P = 2.491 \cdot 10^{-8}$, Supplementary Data 2). In a plant population where seed production is pollen limited, plants will compete for access to pollinators and thus attractiveness to pollinators is a key fitness component. For these assays, we used bumblebee workers (*Bombus terrestris*). Bees were purchased in Switzerland (Andermatt Biocontrol Suisse AG). Hives were kept in a flight cage (75x75x115 cm). When bumblebees arrived, they did not have any experience with plants, therefore, we fed them on *Brassica rapa* "fast cycling" plants for at least a week so they could gain experience with these flowers. Supplemental pollen (Biorex, Ebnat-Kappel, Switzerland) and sugar water (Biogluc sugar solution, Biobest) were also provided for feeding. "Feeding plants" were non-evolved plants and grown on standardized soil to avoid inducing bias in pollinator choice for any of the evolutionary lines. Three days before choice tests and pollination, we removed all plants from the cages, and bumblebees were only fed on supplemental pollen and nectar solution. To enhance bumblebee's foraging activity, bees were starved 16 h before choice tests.

Preliminary assays with our experimental plants had shown that bees generally have a strong preference for non-herbivory plants that had evolved in tuff soil, compared to plants that evolved with herbivory or/and in limestone. This preference was likely due to the bigger size and/or greater flower display of plants that evolved in tuff without herbivory[39]. Therefore, we performed the choice tests with plants in limestone or tuff, as well as those that evolved with herbivory or without separately, to allow for a balanced comparison of attractiveness between treatments. In summary, for each replicate within treatments, we assayed separately four different test groups, containing 16 plants in each assay: (i) plants grown in limestone soil that evolved with herbivory, (ii) plants in limestone soil that evolved without herbivory, (iii) plants in tuff soil that evolved with herbivory, iv) plants in tuff soil that evolved without herbivory. For every test group, the 16 plants were growing in the same soil (either limestone or tuff soil); eight of those plants had evolved in the given soil type and eight plants had evolved in the other soil type. Of these eight plants, four had evolved with bee-pollination and four with hand-pollination. Thus, each treatment was represented (replicated) with four plants within each test group for a total of 52 to 60 tested plants per treatment group. Each test group was replicated 13 to 15 times, using 78 to 90

bees. In the non-herbivory test groups, we used a total of 90 bumblebees for 240 plants in tuff soil and 82 bumblebees for 224 plants in limestone soil. In the herbivory test groups, we used a total of 78 bumblebees for 208 plants for plants in tuff soil and 90 bumblebees for 240 plants growing in limestone soil.

Choice tests were done on the usual pollination date of the limestone- soil evolutionary lines (30 days after sowing out), between 08.30 am and 05.00 pm, in the greenhouse, in a flight cage (l x w x h: 2.5 m x 1.8 m x 1.2 m) under controlled light and temperature conditions. Plants were placed into the flight cage in a 4 × 4 Latin square design with a distance of 20 cm between plants. To ensure that choices were not due to plant position, we changed the plant positions between test groups, and we also released the bumblebees at different locations. For each test group of 16 plants, a total of six bumblebees were used for first-choices assessment. Bumblebee were individually released and recaptured as soon as the first visit was done; bees were only used once. We considered a first choice when bees landed on an inflorescence and started to collect nectar and/or pollen.

## Plant traits

For plants traits measured, a comprehensive approach was chosen to include as many potentially interesting traits as possible. All plant traits were measured in plants of generation ten, and all at the same time to minimize variation due to differences in plant developmental stage. Plant and flower measurements were done within the four days before choice tests (day 30). For measuring plant floral morphology (petal display: petal width, petal length, petal area, sepal length, flower diameter; reproductive organs: long stamen length, pistil length), we sampled three flowers per plant; we placed the floral parts on a white paper sheet and scanned the sheets. Floral traits were quantified from the pictures using the software package Image J. For phenotypic analyses, we calculated the mean value for each trait from the three sampled flowers. Other traits such as the number of leaves, length, and width of the first true leaf, number of open flowers (at day 30), flower production (number of flowers produced from the start of flowering to day 30), number of branches, plant height at day 30, length of branches were also recorded on the same day as choice tests. Leaf size was estimated by multiplying the leaf width by the leaf length and then dividing by two.

## Sequencing and bioinformatics

After phenotypic measurements, fresh leaf tissue of 1118 plant individuals from all treatment groups (min. 20, max. 37 individuals per treatment and replicate) was sampled for genotyping, and frozen at −80 °C. DNA extraction, resequencing, and initial quality control was done by BGI (Shenzhen, China), after which 20Tb of trimmed, clean raw reads for further processing were delivered. Fastqc/multiqc was applied for individual quality control of fasta files. Alignment of fasta files was done using the Burrows-Wheeler algorithm (BWA; v0.7.17[64];) against the Chiifu (v3.0) reference genome[65] (BRAD, http://brassicadb.cn). For variant calling, we used tools from GATK4[66]. Variants were called for each accession individually with HaplotypeCaller using default parameters with one exception: MappingQualityRankSumTest and ReadPosRankSumTest were calculated in addition to the standard attributes. The output GVCF files were combined and consolidated using GenomicsDBImport and joint genotyping was performed with GenotypeGVCFs. In the absence of a high-quality variant set, we took the suggested hard-filtering thresholds and slightly adjusted them for our data set (QD < 5.0, MQ < 40.0, FS > 60.0 SOR > 3.0, MQRanKSum < −5.0). Further, we filtered SNPs with VCFtools (v0.1.15) as follows. We removed all indels and kept only biallelic SNPs with quality above 20 and depth between 6 to 100; the minimum allele frequency (maf) was set to 0.01. After this filtering, 6'277'185 SNPs were kept for all samples in the downstream analyses. The mean ± s.d. depths for all sites and individuals was 41.2 ± 11.3 (min: 16, max: 102).

## Statistical analysis

**Traits associated with fitness.** Seed sets could not be assessed directly in our experiments, because samples for DNA analyses were collected from the plants after the bioassays, leaving some of the smaller plants (especially those growing in limestone soil) with a significant proportion of their biomass removed. Also, accounting for maternal effects in seed production in plants growing in foreign soil would have taken a two-generation approach which was not feasible in this study. To assess which traits were most closely associated with fitness we ran a generalized linear model with (i) relative seed set and (ii) bumblebee first choices as dependent variables, replicate as random factor, and all measured traits as covariates. We used data for the relative seed set, assessed in plants of generations one, two, and seven, to evaluate associations between plant fitness and plant traits across the evolution experiment[39]. Bumblebee's first choices was taken as a measure of the attractiveness of plants to these highly efficient pollinators[59]. Bumblebee visitations of generations one and two showed that the number of visits is highly correlated with seed set in this obligatorily outcrossing, animal-pollinated plant ($r_{784} = 0.62$ P = 2.491*10$^{-83}$, Supplementary Data 2), and thus is a good fitness proxy. Because the "relative seed set" had a zero-inflated distribution (ca. half of the plants in each replicate did not produce any seeds), we used two separate models: a binary response model (with seeds/no seeds as binary response variable) and a linear "truncated regression" model (including only plants that produced seeds; see[39]; Supplementary Tables 1 and 2, Supplementary Data 2). The trait most strongly and consistently explaining attractiveness to pollinators and relative seed set was the "number of open flowers" (Supplementary Tables 1 and 2 and Supplementary Data 2), which was subsequently used to assess local adaptation. Number of open flowers is a trait typically correlated to fitness and has been used as fitness proxy in plant studies of plant local adaptation[6,7,10,11]. Although we have no data to show that the number of open flowers at "pollination day" is positively associated with seed set when plants produce flowers and are visited by pollinators over the whole flowering season (i.e., the most natural setting), our study shows how a trait under selection and highly associated with fitness in this particular experiment, shows GxE interactions consistent with local adaptation. Thus, even if the number of open flowers on a particular day (i.e., day of pollination) may not be the ideal fitness measure in field studies, it serves as a good fitness proxy for proof of concept. In our dataset, the "number of open flowers" on pollination day is not significantly correlated with flowering time (Pearson, $r_{1355} = −0.05$, $P = 0.082$, Supplementary Data 1); thus, more flowers on the day of pollination in our study is not merely a function of earlier flowering, but an investment in rapid flower production, a trait that is under selection in our study.

**Trait evolution.** Evolutionary changes and the impact of soil, herbivory, pollination, and their interactions were evaluated in plants of generation ten. This analysis was only done for individuals growing in their "local" soil. We used a linear mixed model (package lme4[67];), with individual traits as dependent variables, replicate as a random factor and "soil line", "pollination", "herbivory" and their interactions as fixed factors (Supplementary Table 3 and Supplementary Data 1). To simplify the interpretation of significant interactions, we also examined the evolutionary changes within each soil line independently (Supplementary Table 4 and Supplementary Data 1). Values of individual traits were also compared between plants of generation one and ten, for plants growing in the same soil using pairwise post hoc comparisons (see below; Fig. 1; Supplementary Tables 6−9 and Supplementary Data 1).

**Local adaptation.** We ran a general linear model with a normal distribution (SPSS), using number of open flowers (and all other traits) as response variable, soil (the soil the plants were grown in), soil lines (the

soil the plants evolved in), biotic treatment (NHH: no-herbivory hand-pollination, HH: herbivory hand-pollination, NHB: no-herbivory bee-pollination, HB: herbivory bee-pollination), as well as the interactions soil x soil lines, and soil x soil lines x biotic treatment as fixed factors, and replicate as a random factor (Supplementary Table 5 and Supplementary Data 1). Subsequently, because of the significance of the three-way interaction (soil x soil lines x biotic treatment) for "number of flowers", potentially indicating local adaptation differs between biotic treatment groups, we analyzed the biotic treatment groups separately (Supplementary Tables 1, 6–9 and Supplementary Data 1). In these subsequent analyses, a significant soil lines x soil (i.e., G x E) interaction was considered as an indication of local adaptation and was analyzed further by using the linear contrasts of the glmm as post hoc tests (lsmeans function: package emmeans). Linear contrasts were used to compare the different treatments groups following the criteria of local adaptation: "home" versus "away" and "local" versus "foreign"[3]. For local vs foreign, we compared treatment groups where plants of one soil line grew in "their" soil, with treatment groups where plants of the other soil line grew in the same soil. For home vs away, we compared treatment groups where plants of one soil line grew in "their" soil, with treatment groups where plants of the same soil line grew in the other soil. Following[3] the "local" versus "foreign" contrast is the more appropriate criterium for local adaptation because this criterium provides information on the efficacy of divergent selection between populations and therefore on local adaptation. A significant "home" versus "away" contrast on the other hand, is considered to offer supplementary support for local adaptation.

Bumblebee's first choices were analyzed using a generalized linear model with Poisson distribution, with "first choices" as the dependent variable, and soil lines and replicate as random factor. These analyses were done separately for each treatment group and for soil, as the bioassays were conducted separately for soil and herbivory groups.

All trait values of plants of generations one and ten, and the effects of soil lines and soil were also reported and analyzed separately for the herbivory/pollination treatment groups using a linear mixed model (package lme4) with individual traits as dependent variable, replicate as random factor and treatment as a fixed factor, followed by pairwise post hoc comparisons between plants of generation one and plants of generation ten of the different treatment groups (these comparisons were only done among plants growing in the same soil; Supplementary Tables 6–9 and Supplementary Data 1). We chose not to apply Bonferroni correction of *P*-values because our analyses focus on one trait, "the number of flowers", rather than trait combinations (i.e., table-wide statistical differences).

**Genomic analyses.** Allele frequency (AF) change along the genome was calculated between generation one and ten, for each treatment (*n* = 8) and for both replicates separately (i.e., 16 times) by subtracting allele frequencies of generation one from allele frequencies of generation ten. The significance of AF change was estimated by employing the Cochran–Mantel–Haenszel (CMH) test with FDR correction across replicates within treatments[68], where significant sites were those with a *P*-value < 0.05. This method has been shown to have high power and a low false positive rate when compared to other methods[69]. To correct for linkage, SNPs within 2 kb windows were pooled, and only the SNP with the lowest p-value within each window was used. A window size of 2 kb was chosen according to previous linkage decay analyses[52]. The qqman package was used to create Manhattan plots. SNPs with significant values of AF changes in both replicates per treatment group were then used to calculate the mean AF change for each treatment group and paired for the two soil types. SNPs that had a significant value of AF change for only one replicate, or in only one soil type were discarded.

For estimation of the association of SNPs with the trait "the number of open flowers", we used genomic prediction with the R

package rrBLUP to calculate breeding values for SNPs[70]. To do so, a genome matrix with −1, 0, 1 values was extracted from the respective VCF files, where -1 and 1 represent the two allelic states of a marker, and 0 represents the heterozygous state, using the --012 function in VCFtools[71]. The A.mat function in the rrBLUP package was enabled to replace missing markers with the population mean for this marker, and to remove markers with more than 50% missing data. To estimate breeding values of markers, a ridge regression model was used: $Y = \mu + Xg + e$, where $Y$ = Z-transformed values of numbers of flowers, $\mu$ = mean of the training set, $X$ = the genotype matrix containing all markers, $g$ =;the marker effect matrix that is calculated by the model, and e = a vector of residual effects. Sixty percent of cases of each data set was set a training set for the BLUP model. The model was run for each treatment and replicate separately, including plants grown in local and foreign soil, but without including data from generation one (i.e., 16 times with a sample size of between 45 and 72 individuals each). SNPs with a marker effect in only one replicate were discarded. The resulting matrix of SNP effects for each treatment and replicate was matched with the matrix of markers with significant allele frequency change, so only markers with significant allele frequency change and a marker effect value from the genomic prediction in both replicates were used in the analysis.

To estimate patterns of adaptive response in these markers, the AF changes were analyzed for the two soil types in the four pollination/herbivory treatments. The following criteria were set for two different evolutionary scenarios: (1) antagonistic pleiotropy (AP): allele frequency change greater than 0.1, in opposite directions in the two soil types; (2) conditional neutrality: allele frequency change of greater than 0.2 in one soil, in either direction, but less than 0.1 in the other soil type. These threshold values (i.e., absolute AF > 0.2) were assumed following previous studies done in the field[7]. Only SNPs that fulfilled these criteria in both replicates were considered consistent in their evolutionary pattern and thus included in the final analysis of the commonness of the evolutionary scenarios in the different treatment groups. Because these are conservative criteria, we suspect that false negatives for both AP and CN are fairly common. Also, we cannot rule out false positives for CN, since some neutral alleles may be under weak selection. Nonetheless, these criteria offer a robust means for identifying general differences in the genomic architecture of selective responses in the different treatments.

To identify genes underlying the SNPs belonging to one of these scenarios, we retrieved genes around the significant markers. We selected genes matching the marker or in the surrounding region (1 kb upstream and 1 kb downstream) using a custom script (Supplementary Data 3–5). We used the annotation features from the reference genome Chiifu (v3.0[65],) available on NCBI. To analyze the commonness of AP in the different biotic treatments, a binary generalized linear model was used, with the presence of markers with an AP pattern as the response variable, and treatment (HB, NHB) as a factor. Statistical analyses were performed with R-Studio software 4.0.0 (2020, R Foundation for Statistical Computing, Vienna, Austria) and IBM SPSS statistics.

**Discriminating drift from selection.** We used multiple approaches to control for the potential effects of drift in our experiment. To discriminate the effects of natural selection from drift in the phenotypic data, we assessed whether trait differences were consistent among replicates of a given treatment group. In the GLM analysis, a significant "treatment" effect indicates trait differences between different treatment groups across the replicates. Drift would be indicated by evolutionary changes in one replicate only, indicated by a significance in the factor "replicate". At the genomic level, drift was controlled for by using the CMH- test with False Discovery Rate correction of *P*-values. This test draws power from treatment replicates to test for consistent changes in AF, that are unlikely due to genetic drift after controlling for multiple testing. While FDR-corrected CMH inference identifies loci

with AF-change likely caused by selection, for testing the role of antagonistic pleiotropy versus conditional neutrality, we further sub-setted these loci to analyze just those with the strongest link to fitness - loci with consistent AF change across replicates that are also associated with number of flowers. More generally, our study also draws power out of its comparative nature. Because factors that may impact random change such as pollen limitation, effective population size, and generation time were identical/similar across treatment groups, the effects of drift should also have been very similar in magnitude, and differences between treatments be mostly caused by selection.

### Reporting summary

Further information on research design is available in the Nature Portfolio Reporting Summary linked to this article.

## Data availability

Phenotypic data are available on dryad: https://doi.org/10.5061/dryad. k98sf7mfm. Genomic sequences analyzed in this article are stored and accessible through the National Center for Biotechnology Information (NCBI) under https://www.ncbi.nlm.nih.gov/bioproject/PRJNA1105729.

## Code availability

R scripts used to analyze the data in this study are available on dryad: https://doi.org/10.5061/dryad.k98sf7mfm.

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

## Acknowledgements

We thank Giovanni Scopece, Salvatore Cozzolino, Laura Dällenbach, Franz Huber, Rayko Jonas, Markus Meierhofer, and Luca Arrigo for their invaluable help with this study. Tyler Figueira provided helpful advice on R-coding and statistical testing and Roman Briskine helped with the bioinformatic analyses. Tadeusz J. Kawecki provided helpful insight on the design of this experiment and early analyses. Judith Bronstein and Sharon Strauss provided helpful comments to an earlier version of the manuscript. The research was funded by the Swiss National Science Funds (SNF grant no. 31003A_172988 to FPS).

## Author contributions

Conceptualization: F.P.S., T.D. Experimental: T.D. Analysis: T.D, F.P.S, L.F., L.H.R., and J.M.K. Writing: F.P.S., T.D., L.H.R., L.F., and J.M.K.

## Competing interests

The authors declare no competing interests.
