## [Peer Review File · Nature Communications]

Biotic interactions promote local adaptation to soil in plantsREVIEWER COMMENTS

Reviewer #1 (Remarks to the Author):

In this paper, the authors attempt to analyse with experimental evolution how three environmental factors (soil type, herbivory, and pollination mechanism) may interact to shape adaptive evolution. In the analysis, the focus is on adaptation to soil and how this might be affected by selection mediated by bumblebee pollinators, and by selection mediated by aphids.

The paper addresses a problem of general interest using an experimental system in the lab. The design is complex and the presentation is not always easy to follow. The authors point out that experimental conditions were not designed to mimic those of any particular field environment. Still, I think it would be of interest to discuss the extent to which results are likely to be a function of the specific experimental procedures used, and the extent to which they are likely to apply also to natural populations in the field.

I have four main comments regarding conceptual framework, methods and statistical analysis.

First, I suggest the authors should attempt to formulate some predictions in the Introduction. This would help the reader sort through the many results presented. For example, why were the 12 traits quantified expected to be influenced by the experimental treatments, and be important for adaptation to soil conditions?

In the test for local adaptation in generation 10, plants were grown without herbivores and were subject to hand-pollination (p. 10, line 2). Wouldn't an a-priori prediction be that under these conditions, replicates that had been exposed to this treatment combination on the respective soil during experimental evolution would do best? The authors instead find that in the fitness assay in generation 10, replicates subject to the combination of tuff soil, aphid herbivory, and bumblebee pollination during experimental evolution perform best on tuff soil, and the replicates subject to limestone soil, aphid herbivory, and hand-pollination do best on limestone (Fig. 2). In the Discussion the authors suggest that the reason why a trade-off between adaptation to limestone and tuff is detected only in the treatment involving both herbivory and bumblebee pollination could be because "biotic interactions reinforce trade-offs caused by adaptation to different soil types" (p. 6, line 3). This is an interesting idea, but would it not also imply that there is no fitness cost to adaptation to these pollinators and herbivores (a cost that would be expected to be expressed in environments where these interactions are not present)? Could this be related to the way fitness is quantified in this experiment? The authors may want to comment on this in the Discussion.

Second, for studies of local adaptation it is critical that relevant measures of fitness are used. In the present paper, the authors use number of open flowers at day 30 and the seed set of these flowers to determine fitness. The "relative seed set" of these flowers is used to determine the contribution of individual plants to the next generation in each treatment in the experimental evolution. The authors also report that seed set is correlated with bumble visitation and suggest that also bumblebee visitation preference therefore can be used as a fitness measure (p. 12, line 6).

Do the authors have any additional data to show that number of open flowers on day 30 is a good measure of total flower, fruit, or seed production across the season when the study species is grown on the soil types used in the present study for a period representing a "typically long" growing season for this species? Or is the selection applied based on this criterion likely to result in selection for producing many open flowers at this particular point of plant development, rather than a high total flower and seed output across the whole growing season? This is of interest in relation to the interpretation of results. Is the selection applied likely to some extent reflect variation in selection in the field, or should this be considered a fully artificial situation, with the goal only to demonstrate effects "in principle"? I suggest the authors in the Discussion consider the specifics of experimental conditions and the selection regime applied in relation to what they might be in natural populations of this species.

Third, the study is based on a complicated experiment, and the description of its design and analysis is difficult to follow in several places (see comments on details below). The authors should work hard to describe and motivate better and more clearly the experimental procedures used and the various analyses conducted.

With three treatments crossed in the experimental evolution (soil, herbivory, and pollination treatment), there are many possible perspectives to take on interactions among treatments. The authors have decided to focus on adaptation to soil. Any reason for not also considering adaptations to herbivory and to pollination regime in this experiment, and how they are affected by soil conditions?

Fourth, I am not convinced that the correct statistical model has been used to analyse the effects of selection regime on trait differentiation over time and local adaptation (Tables S3 and S4). It seems that in these analyses observations of individual plants have been considered independent observations of the effects on phenotypes of treatment combinations (combinations of soil, herbivory and pollination treatments in the experimental evolution). However, plants within a replicate of a given treatment combination are not statistically independent. There were only two replicates of each treatment combination in the experimental evolution, and this should be reflected in the statistical model and significance testing. Wouldn't it be more correct to conduct these analyses (Tables S3 and S4) on replicate mean values (where replicate refers to each of the two independent selection lines of each treatment combination)?

In addition, it is not clear why the authors have chosen a model that does not take the full factorial design into account when estimating effects of treatment on fitness and trait expression in generation 10. Instead of including soil, herbivory and pollination treatment in the experimental evolution as independent variables in the analysis, the authors have opted to combine herbivory and pollination treatment into something called "Biotic treatment". Please, motivate.

Comments on details

p. 1, line 12, "parameters". Change to "factors" here and elsewhere?

p. 1, line 16. Two- and three-way interactions can always be considered with different perspectives. However, it is a bit confusing when the perspective changes from one sentence to the next: "...to study how biotic interactions affect local adaptation to soil. We show that several plant traits evolved in response to biotic interactions in a soil-specific way." The latter sentence seems to indicate that adaptation to herbivory and pollination evolved in soil-specific ways, but was adaptation to soil affected?

p. 1, line 19, "local adaptation evolved in the highly fitness-relevant trait". This statement is unclear for a at least a couple of reasons.

First, what is a "fitness-relevant trait"? The definition of local adaptation is based on measures of fitness. Is number of open flowers here considered a measure of fitness, or a trait likely to affect fitness? If the former, perhaps rephrase to say that local adaptation was quantified based on number of open flowers 30 days after sowing? If the latter, the statement would rather be understood such that differences in number of open flowers reflected differences in optimal number of open flowers in different treatments. Please, clarify.

Second, should this sentence be understood such that local adaptation to soil only evolved in one combination of herbivory and pollination treatments, or that only this combination of treatments showed an adaptive response compared to other treatments? I suggest this can be clarified by specifying to what environmental condition local adaptation evolved (to soil?).

p. 1, line 24, "Our results...". Any indication of the reason behind this result?

p. 1, line 45. Shouldn't mycorrhizal mutualists be sorted under "biotic factors" rather than "chemical parameters"?

p. 2, line 11, "the hypothesis that biotic interactions by changing patterns of selection". Is it

possible to formulate some more specific predictions regarding the expected effects of the tested biotic interactions on phenotypic evolution, and how this may affect plant adaptation to soil?

p. 2, line 17, "followed by two generations without insects". Why were treatments changed the two final generations? Please, motivate. Does this mean that all plants in all treatments were pollinated by hand the two final generations?

p. 2, line 21. Indicate why these particular traits were chosen for study.

p.2, line 31. Why were only plants growing in their local soil included in the analysis? It would seem natural to include also plants grown in the other soil here.

p. 3, line 23. "Relative seed set" here refers to the seed production by the flowers that were open on "pollination day", if I understand this correctly. It is not surprising that this number is strongly correlated with the number of flowers that were open on that day. Is number of open flowers "on pollination day" likely to be correlated with overall flower and seed production of plants receiving pollination across a growing season, or is number of flowers on that day mainly a function of when a plant begins to flower?

p. 3, line 25. Is biomass removal for DNA sampling likely to have affected the seed quality of small plants more than that of large plants? May this have influenced the relative performance of plants selected for the following generation?

p. 3, line 30. The definition of "soil-line" is unclear and confusing. First, it is not likely to represent a single plant genotype (p. 3, line 30). Several selection lines were evolving on each of the two soil types (2 replicates of each of the four herbivory x pollination treatment combinations), and each selection line can be assumed to have consisted of several genotypes. Would simply "soil" work to indicate soil treatment during the experiment, and then use an expression such "soil environment" for the soil in which the different selection lines were grown to test for local adaptation in generation 10?

p. 3, line 41, "for "time to first flower", the performance pattern matched the local vs. foreign criterion"". In my view, local adaptation can only be assessed with estimates of fitness. Whether a given pattern of trait variation can be expected to contribute to local adaptation should depend on the direction of selection on this trait in the environments examined.

p. 4, line 14, "there was no overlap in the genes associated to AP-markers in the two treatment groups". This is an interesting observation. Can it be compared to how consistent patterns were across replicates of individual treatments? How often were allele frequency changes in opposite directions recorded in only one of the replicates of the two treatments compared?

p. 4, line 18, "No pattern matching..." Please, explain what this means and which results motivates this conclusion. How does this statement square with the legend to Fig. 2, second last sentence, where "conditional (CN) in both replicates" is mentioned?

p. 5 line 10, "markers with breeding values for "number of open flowers"" Please, explain what this means.

p. 5, line 31, "either herbivory or pollination for many traits (Table S4)". Why not include the factors herbivory and pollination in the analysis presented in Table S4, rather than the factor "Biotic treatment"? This would allow the authors to separate the effects of the two treatments.

p. 3, line 3. Wouldn't a priori the strongest evidence of local adaptation had been expected in the "NHH" treatment, where conditions during experimental evolution were likely to be the most similar to those of the "reciprocal transplant experiment" (cf. Fig. 2)? The authors argue that biotic interactions may reinforce trade-offs caused by adaptation to different soil types. At the same time, the results apparently suggest that adaptation to the biotic agents is not associated with a fitness cost in an environment where they are not present.

p. 6, line 35, "traits showing a pattern of local adaptation". Unclear what this means. A significant origin x environment interaction ("soil x soil-line interaction" in this case) for an arbitrary trait does not necessarily indicate local adaptation. For such an interaction to indicate local adaptation would require that in each environment, the "local genotype" is characterized by a phenotype closer to the optimal value in that environment. In other words, data on phenotypes are not sufficient; in addition, something needs to be known about the optimal phenotype in the contrasting environments.

Methods

p. 7, line 22. Were plants grown in individual pots, or in experimental arrays with 49 plants in each? Were plants of a given treatment kept together throughout the experiment, or were they fully randomized except during bee-pollination and herbivory treatments?

p. 7, line 25, "Plants grown from the seeds of these different crosses..." Unclear what the message of this sentence is. Can be read as if there were no phenotypic differences among treatments, which apparently there were according to the results.

p. 8, line 13, "to provide overall insights". Unclear what this means.

p. 8, line 15, "and support concepts of how different ecological parameters interact to drive plant evolution". The authors may want to specify which concepts they have in mind.

p. 9, line 4. Pollination was allowed on a single day (10 days after herbivory). At this time not all plants were flowering, and among those flowering, flower number varied. This should result in strong selection on flowering phenology, and more specifically on high flower production early in the season. Are there any data indicating that early flowering start is correlated with high total flower and fruit production in *Brassica rapa* when grown on these substrates also when the growing season is not artificially short? Previous work on the rapid-cycling population has indicated that late flowering start may be associated with high total reproductive success (Agren and Schemske. 1993. *Am Nat*), and also that optimal flowering time will depend on environmental conditions (Johnson, Hamman, Franks. 2022. *AJB*).

p. 9 line 28. Explain abbreviations used for the different treatments.

p. 9, line 22. Please, clarify procedure. Was the pollen donor randomly chosen among the 28 plants assigned as pollen recipients in the hand-pollination treatment, or was it randomly chosen among all 49 plants in the replicate?

p. 10, line 2. What was the design of the hand-pollination treatment in generation 9 and 10? Were all plants and flowers pollinated by hand this time?

p. 10, line 6. How were seed families selected for inclusion in the reciprocal transplant experiment (40 of 49 families included)? Did in this case all families in a replicate have an equal chance to be included (unlike during the experimental evolution)? If not, the change in pollination treatment in some treatments in generation 9 might influence composition of generation 10.

p. 10, line 15, "*Brassica rapa* is an outcrossing species and therefore its reproductive success is highly correlated to pollinator visitation..." Note that such a correlation does not necessarily reflect variation in mating success, as might be subsumed based on the mentioning of pollinator visitation. It could arise because pollinator visitation is negatively correlated with pollen limitation of seed production. However, it could also arise because large plants with many flowers are more attractive to bumblebees without the need of any variation in pollen limitation.

p. 10, line 31 and onwards. The description of the bumblebee choice experiments is tough reading. Would a graph perhaps help explain the various set-ups?

I cannot see that this experiment and its analysis (p. 12) take into account the non-independence of plants that are the product a given replicate of the treatments in the experimental evolution. Will this not result inflated degrees of freedom for the test of treatment effect? The study is set up to examine whether conditions during experimental evolution affects evolutionary outcome, and

whether this outcome is consistent across replicates. Shouldn't replicates rather than individual plants be considered independent observations of the effects of different selection environments?

p. 10, line 33. Does this describe phenotypic differences observed between treatments during the experimental evolution, as well as differences observed between treatments when grown under common conditions in generation 10?

p. 10, line 37, "each replicate". Does this refer to replicates in the experimental evolution, or to replicates of the bee-choice assay?

p. 11, line 11. The authors should motivate why these particular traits were chosen for study. Did the authors have any specific predictions regarding evolutionary response to the different selection regimes? In particular, how were these traits expected to influence performance on the two different soil types?

p. 11, line 46. Does this describe the analysis of traits scored in generation 10, or something in addition to that? Please, clarify.

p. 12, line 2, "relative seed set" as dependent variable. Confusing. From the description above, one gets the impression that seed set was not scored, but only bumble first choices. Please, clarify.

p. 12, line 3, "all measured traits". Please, specify which they were. Twelve traits are listed in Tables S3 and S4. To examine the independent effect of all of these traits on fitness measures in replicates that consisted of 49 plants does not seem meaningful.

p. 12, line 6. A slightly different correlation coefficient is reported on p. 10, line 15.

p. 12, line 24. Why were "all traits" examined in an analysis of local adaptation? Only variance in fitness estimates will provide insight into possible local adaptation.

p. 12, lines 30-33. The authors may want to reword this to indicate that a significant interaction with soil does not by itself indicate "local adaptation to soil". Such an interaction would need to be associated with a higher relative fitness of the "home" genotype than of the "away" genotype on both soils, which is tested in the contrasts described later.

Table S3, table head, "Plants that evolved in the different soil types were analysed separately, because the analysis contained only plants growing in their native soil type..." This statement is difficult to understand, and should be rephrased. I think it is the word "because" that makes the statement confusing.

Reviewer #2 (Remarks to the Author):

The authors address an important and complex question regarding the interaction of abiotic and biotic factors as drivers of plant adaptive evolution. Using a manipulation experiment run over multiple generation with a factorial design for soil, herbivores and pollination, they found that traits of *B. rapa* evolved in response to different selection regimes, and that biotic interactions promoted adaptation to soil. They further dissect the genetic basis of this adaptation process by analysing temporal allele frequency shifts and associating candidate SNPs to genetic trade-offs. The study relies on an impressive set of experiments conducted over multiple generations, which, coupled with trait measurements and sequencing, make for a great dataset. Findings are potentially highly interesting and unique in dissecting the role of interactions contributing to evolution, a perspective that is rarely seen in the field. However, the system is complex and results are numerous, which calls for a clear text to allow the reader to follow. The text is currently written in a short format (I guess shorter than the requirements from Nature Communications) but in my opinion, more information and clarity are needed to make the content accessible. Some of the key conceptual statements that underly the study should also be revised. The manuscript could

present a highly relevant contribution following revision.

The manuscript aims to address the role of biotic and abiotic factors in driving adaptation, but which evolutionary response is the focus of the study is unclear. In the introduction, the problem is pitched on the role of soil (abiotic) in driving the response to selection imposed by herbivores and pollinators (biotic; P2L1-7). However, later the main question seems to be how biotic factors impact adaptation to soil (P2L11-12). The repeated switch between these two perspectives in the manuscript makes it difficult to follow. I expand on these perspectives below.

The experimental evolution analyses (Fig1, TableS3) look at how selection imposed by biotic factors drives phenotypic shifts, and while experiments are run on separate soils and there is no test of soil x biotic interactions, the text keeps emphasising how the evolutionary responses vary (without statistical evidence) across soils. I do not understand while soil interaction is not tested, as this seems to be a crucial point, not just because the text hints at it repeatedly and the way Fig1 is structured (showing contrasts across soils), but because it is at the basis of the factorial design that is emphasised in the methods. In this regards, I do not understand the statement at P2L31-33. One could guess that many soil interactions are significant, such as the example reported for opposite pollination effects on "Time to first flower" (P2L25; Table S3). Overall, my take home message of this experiment is that soil mediates evolutionary responses to biotic factors (though again, there is no formal test for this, but this is what I deduce from the text). Next, in the transplant experiment (Fig2), the agent of selection switches from biotic factors to soil, and the focus is now on soil adaptation as mediated by biotic factors. Indeed, this fits what is stated as the main goal at P2L10-122. Results here are interesting, but their interpretation in the discussion is only quickly touched upon. Most of the arguments related to this experiment are instead in line with the perspective of the first experiment (e.g. P5L32 [but not tested], P5L42 and 43; P6L5-8). Overall, I think the misalignment of hypotheses, experiments, and interpretations across the manuscript leaves the reader wondering what question is actually addressed. I realise that dissecting the role of these interaction is complex, but the way this is implemented in the analyses and in the text does not seem coherent, and the text could be improved to sharpen the focus.

Genomic analyses of allele frequency shifts and genetic trade-offs are very interesting but I think more explanations of the results are necessary. The manhattan plots show significant outliers across the whole genome, which calls for a plausible explanation to rule out the suspect of a large number of false positives. Some regions seem to have a clearer signal than others, as identified by discrete peaks emerging from what seems to be a noisy background signature. I understand that initial commercial lines were genetically diverse and plants are self-incompatible, so I imagine LD is low in this system? I guess SNP signal can be very noisy and a window analysis may yield more discrete evidence. You could still take SNPs within significant windows for the AP vs CN downstream tests. Alternatively, a neutral expectation through simulations may be needed here.

The second part of the genomic analyses relies on the BLUP estimate of SNP fitness effects. More information is needed in the methods to assess how the analyses were run and significance of the results. Why was a training set used, isn't the number of flowers available for all individuals? I can't understand how the sample size for each treatment and replicate can be > 49? Does it mean that predictions were run on 45-72 samples independently 16 times, and how were these data combined to give values plotted in Fig3? This seems a very low sample size for such predictions. Effects can certainly have high values but also very large SE. Was any filtering for significance applied? A table with effect, SE and significance should be provided. It would also be very interesting to see these effects combined with the manhattan plot. This could address one of your statements in the introduction (P1L36) of unclear links between genetic trade-offs and divergence. I guess the expectation would be that strong divergence should emerge from AP effects?

Figure 1

As mentioned above, Fig1 is misleading and I find it difficult to match to the text. In comparison, Table S3 is easier to look at. Perhaps you could consider combining plots and statistics? I think plots would be clearer separated by soil (instead of combined).

What are the bars on top of each column?

Sum of visits is not reported in TableS3. Bars are missing for these columns.

As far as I can figure use of letters for significance relies on fixed effects of TableS3, so their use is different from what is commonly seen, which is individual contrasts of all values represented in the plot. For example, for Number of flowers in Tuff, BB seems to indicate that two aphid treatments are not significant, while they are significant from columns indicated with A. I also don't understand why G1 receives a letter and results from this contrast are not presented in the text. Please clarify.

Figure 2

How are manhattan plots generated? There are 2 replicates analysed independently, so what is represented here?

P1L36-40: many concepts in 4 lines without explanation. Some of these do not seem relevant for this study as they are not further addressed (e.g. ecological speciation).

P1L32+38: clarify these statements relevant for the manuscript, we do know something about these processes

P1L36: the link between antagonistic selection and AP vs CN is not clear

P1L46: report what is known about adaptation to pollinators and herbivores, or on the clarified study questions

P2L20: 'native' soil is not defined at this point as it comes only in the methods

P2L11-20: introduction of traits or fitness measured is needed to be able to follow results

P2L21: what's the evidence for resource reallocation?

P2L24-40: descriptions difficult to follow

P3L15: reciprocal transplant experiment has not been presented before

P3L36: results difficult to follow

P4L4: pvalue missing

P4L19: what does this mean: "pattern that matches phenotypic data"?

P5L32: this is not tested

P12L38-44: home vs away is not reported in results

RESPONSE TO REVIEWERS' COMMENTS

Dear reviewers,

We would like to express our sincere gratitude for your constructive and extensive comments in attempting to improve the manuscript. We would like to highlight that some of the comments emerged from the unusual structure of the manuscript, that was originally prepared for "Nature" and then, without change in format, transferred to "Nature Communications". We have now changed the structure of the paper according to the guidelines of Nature Communications, which led to major restructuring, and addresses many of the comments that refer to lack of clarity in the description of methods and results. The introduction to the re-structured manuscript now includes a methods paragraph, an explicit description of the hypotheses tested, and a motivation for the main aspects of the experimental design. Some more detailed aspects (such as further explanations on the fitness measure) were moved into the methods section. In the results section, the phenotypic evolution and local adaptation analyses have been moved together.

Reviewer #1 (Remarks to the Author):

In this paper, the authors attempt to analyse with experimental evolution how three environmental factors (soil type, herbivory, and pollination mechanism) may interact to shape adaptive evolution. In the analysis, the focus is on adaptation to soil and how this might be affected by selection mediated by bumblebee pollinators, and by selection mediated by aphids.

The paper addresses a problem of general interest using an experimental system in the lab. The design is complex and the presentation is not always easy to follow. The authors point out that experimental conditions were not designed to mimic those of any particular field environment. Still, I think it would be of interest to discuss the extent to which results are likely to be a function of the specific experimental procedures used, and the extent to which they are likely to apply also to natural populations in the field.

I have four main comments regarding conceptual framework, methods and statistical analysis.

First, I suggest the authors should attempt to formulate some predictions in the Introduction. This would help the reader sort through the many results presented. For example, why were the 12 traits quantified expected to be influenced by the experimental treatments, and be important for adaptation to soil conditions?

Specific predictions are now included in the modified introduction. For the traits, we tried to use a complementary approach; even though fitness is of key importance in local adaptation, other traits are also of interest and may show interesting evolutionary patterns. Our experiment also included some "discovery" aspects, that are not driven by specific hypothesis.

In the test for local adaptation in generation 10, plants were grown without herbivores and were subject to hand-pollination (p. 10, line 2). Wouldn't an a-priori prediction be that under these conditions, replicates that had been exposed to this treatment combination on the respective soil during experimental evolution would do best? The authors instead find that in the fitness assay in generation 10, replicates subject to the combination of tuff soil, aphid herbivory, and bumblebee pollination during experimental evolution perform best on tuff soil, and the replicates subject to limestone soil, aphid herbivory, and hand-pollination do best on limestone (Fig. 2). In the Discussion the authors suggest that the reason why a trade-off between adaptation to limestone and tuff is detected only in the treatment involving both herbivory and bumblebee pollination could be because "biotic interactions reinforce trade-offs caused by adaptation to different soil types" (p. 6, line 3). This

is an interesting idea, but would it not also imply that there is no fitness cost to adaptation to these pollinators and herbivores (a cost that would be expected to be expressed in environments where these interactions are not present)? Could this be related to the way fitness is quantified in this experiment? The authors may want to comment on this in the Discussion.

This is an interesting point; the idea here is that biotic interaction speeds up local adaptation by imposing strong selection. There may still be costs to adapting to pollinators and herbivores, e.g. in traits like roots or secondary metabolites that we have not quantified. For the treatment without biotic interactions (especially with bee pollination) there may simply not be enough strong selection for rapid adaptation. All in all, we do not know whether the patterns we see are merely temporal, i.e. local adaptation evolved fastest with biotic interaction, but would also evolve in the other treatment group, given more time, or whether there is a real qualitative difference in the treatments with/without local adaptation. We address this now in the discussion more specifically.

Second, for studies of local adaptation it is critical that relevant measures of fitness are used. In the present paper, the authors use number of open flowers at day 30 and the seed set of these flowers to determine fitness. The “relative seed set” of these flowers is used to determine the contribution of individual plants to the next generation in each treatment in the experimental evolution. The authors also report that seed set is correlated with bumble visitation and suggest that also bumblebee visitation preference therefore can be used as a fitness measure (p. 12, line 6).

Do the authors have any additional data to show that number of open flowers on day 30 is a good measure of total flower, fruit, or seed production across the season when the study species is grown on the soil types used in the present study for a period representing a “typically long” growing season for this species? Or is the selection applied based on this criterion likely to result in selection for producing many open flowers at this particular point of plant development, rather than a high total flower and seed output across the whole growing season? This is of interest in relation to the interpretation of results. Is the selection applied likely to some extent reflect variation in selection in the field, or should this be considered a fully artificial situation, with the goal only to demonstrate effects “in principle”? I suggest the authors in the Discussion consider the specifics of experimental conditions and the selection regime applied in relation to what they might be in natural populations of this species.

Unfortunately, we do not have data showing an association between the number of open flowers on “pollination day” and total seed set of plants during the whole flowering season and natural visitation of pollinators. There is some data in the literature that early flowers contribute more to fitness than later ones ¹ and this may also be true for *Brassica rapa*. Otherwise, our data shows as a proof of concept, that patterns of local adaptation for a trait under selection in this particular experiment, is impacted by biotic interaction (as the reviewer mentions). A notion explaining this interesting point was added to the relevant place in the text. P14 L38

Third, the study is based on a complicated experiment, and the description of its design and analysis is difficult to follow in several places (see comments on details below). The authors should work hard to describe and motivate better and more clearly the experimental procedures used and the various analyses conducted.

OK, done.

With three treatments crossed in the experimental evolution (soil, herbivory, and pollination treatment), there are many possible perspectives to take on interactions among treatments. The authors have decided to focus on adaptation to soil. Any reason for not also considering adaptations to herbivory and to pollination regime in this experiment, and how they are affected by soil conditions?

This is an excellent idea, and in fact we have considered a follow-up experiment investigating adaptation to pollinators and herbivores. Soil adaptation is a prominent topic in plant diversification, thus highly relevant for local adaptation studies. Also, soil was the most natural of all used factors, as the soil was not sterilized and collected in natural Brassica populations. These arguments guided our choice of soil adaptation as the focus of this study. This is now more clearly described in the introduction. P2 L38

Fourth, I am not convinced that the correct statistical model has been used to analyse the effects of selection regime on trait differentiation over time and local adaptation (Tables S3 and S4). It seems that in these analyses observations of individual plants have been considered independent observations of the effects on phenotypes of treatment combinations (combinations of soil, herbivory and pollination treatments in the experimental evolution). However, plants within a replicate of a given treatment combination are not statistically independent. There were only two replicates of each treatment combination in the experimental evolution, and this should be reflected in the statistical model and significance testing. Wouldn't it be more correct to conduct these analyses (Tables S3 and S4) on replicate mean values (where replicate refers to each of the two independent selection lines of each treatment combination)?

In the model, replicates are included as random factors, thus there is a control for the number of replicates. In our experimental design, replicates mimic populations that evolve fully independently under (almost) identical ecological conditions. In this context, individuals need to be considered to assess mean values (per replicate) **plus** variance. This is what our ANOVA (i.e. GLM) based approach does. Analyzing effects merely on replicate level (i.e. using mean values of replicates) would disregard variance within replicates. This is important given the fact that selection acts on **individuals**, and values of individuals make up the mean and variance of a population. Evolution happens in populations (i.e. replicates) of individuals with gene-flow and represents itself as a shift in mean trait values within populations. Therefore, evolution of traits should be analyzed among replicates using mean values and variance within a population. For this reason, we do believe we have used the correct approach, by analyzing individuals within replicates and an analysis on replicate level only would ask a different question, i.e. evolution on an above-population level.

In addition, it is not clear why the authors have chosen a model that does not take the full factorial design into account when estimating effects of treatment on fitness and trait expression in generation 10. Instead of including soil, herbivory and pollination treatment in the experimental evolution as independent variables in the analysis, the authors have opted to combine herbivory and pollination treatment into something called "Biotic treatment". Please, motivate.

We now show results of soil, herbivory, pollination, and their interactions (Table S3) for local lines. It is important to highlight that some interactions were excluded from models analyzing bumblebees' first choices, as bioassays were independently conducted for herbivory and non-herbivory plants.

Comments on details

p. 1, line 12, "parameters". Change to "factors" here and elsewhere?

Done P1 L14, P2L18, P2 L32, P6L30, P9L10

p. 1, line 16. Two- and three-way interactions can always be considered with different perspectives. However, it is a bit confusing when the perspective changes from one sentence to the next: "...to study how biotic interactions affect local adaptation to soil. We show that several plant traits evolved in response to biotic interactions in a soil-specific way." The latter sentence seems to indicate that

adaptation to herbivory and pollination evolved in soil-specific ways, but was adaptation to soil affected?

Wording changed now

p. 1, line 19, “local adaptation evolved in the highly fitness-relevant trait”. This statement is unclear for a at least a couple of reasons.

First, what is a “fitness-relevant trait”? The definition of local adaptation is based on measures of fitness. Is number of open flowers here considered a measure of fitness, or a trait likely to affect fitness? If the former, perhaps rephrase to say that local adaptation was quantified based on number of open flowers 30 days after sowing? If the latter, the statement would rather be understood such that differences in number of open flowers reflected differences in optimal number of open flowers in different treatments. Please, clarify.

added “a trait used as fitness proxy”

Second, should this sentence be understood such that local adaptation to soil only evolved in one combination of herbivory and pollination treatments, or that only this combination of treatments showed an adaptive response compared to other treatments? I suggest this can be clarified by specifying to what environmental condition local adaptation evolved (to soil?).

added “to soil type”

p. 1, line 24, “Our results...”. Any indication of the reason behind this result?

Unfortunately, there is no space here to elaborate on the reasons

p. 1, line 45. Shouldn’t mycorrhizal mutualists be sorted under “biotic factors” rather than “chemical parameters”?

Done; P2L20

p. 2, line 11, “the hypothesis that biotic interactions by changing patterns of selection”. Is it possible to formulate some more specific predictions regarding the expected effects of the tested biotic interactions on phenotypic evolution, and how this may affect plant adaptation to soil?

We have added some predictions to the end of the introduction

p. 2, line 17, “followed by two generations without insects”. Why were treatments changed the two final generations? Please, motivate. Does this mean that all plants in all treatments were pollinated by hand the two final generations?

See P12L15. Plants were grown for two generations without insects to reduce maternal effects on seed quality, which otherwise would be affected by aphid-herbivory.

p. 2, line 21. Indicate why these particular traits were chosen for study.

A sentence was added in P13 L30

p.2, line 31. Why were only plants growing in their local soil included in the analysis? It would seem natural to include also plants grown in the other soil here.

This is now explained in the methods paragraph. P3 L4

p. 3, line 23. “Relative seed set” here refers to the seed production by the flowers that were open on “pollination day”, if I understand this correctly. It is not surprising that this number is strongly correlated with the number of flowers that were open on that day. Is number of open flowers “on pollination day” likely to be correlated with overall flower and seed production of plants receiving pollination across a growing season, or is number of flowers on that day mainly a function of when a plant begins to flower?

We have added explanatory sentences at this point. Unfortunately, we do not have data showing that “number of flowers” at pollination day correlates with total seed set under natural pollination in the field. In our dataset, “number of flowers” at pollination day does, however, not really correlate with flowering time ($r_{1355}=-0.047$, $P=0.082$); thus, more flowers at the day of pollination in our study is not merely a function of earlier flowering, but an investment in rapid flower production, a trait that is under selection in our study.

p. 3, line 25. Is biomass removal for DNA sampling likely to have affected the seed quality of small plants more than that of large plants? May this have influenced the relative performance of plants selected for the following generation?

DNA collection of first and last generations were performed at the end of the experiment (i.e. as in a resurrection experiment); thus, there was no following generation that could have been affected by DNA sampling.

p. 3, line 30. The definition of “soil-line” is unclear and confusing. First, it is not likely to represent a single plant genotype (p. 3, line 30). Several selection lines were evolving on each of the two soil types (2 replicates of each of the four herbivory x pollination treatment combinations), and each selection line can be assumed to have consisted of several genotypes. Would simply “soil” work to indicate soil treatment during the experiment, and then use an expression such “soil environment” for the soil in which the different selection lines were grown to test for local adaptation in generation 10?

We have changed this to “soil lines”, taking into account that soil genotype typically included more than one biotic interaction line.

p. 3, line 41, “for “time to first flower”, the performance pattern matched the local vs. foreign criterion””. In my view, local adaptation can only be assessed with estimates of fitness. Whether a given pattern of trait variation can be expected to contribute to local adaptation should depend on the direction of selection on this trait in the environments examined.

True, it is the fitness that is used to measure local adaptation, but fitness is determined by traits. For most traits, however, we have no proof that they evolve under adaptive evolution. Thus, we have added a cautionary statement highlighting this fact. P4 L38

p. 4, line 14, “there was no overlap in the genes associated to AP-markers in the two treatment groups”. This is an interesting observation. Can it be compared to how consistent patterns were across replicates of individual treatments? How often were allele frequency changes in opposite directions recorded in only one of the replicates of the two treatments compared?

In fact, we have employed a fairly strict criterion for antagonistic pleiotropy as we have only considered loci where the pattern in both replicates were consistent with antagonistic pleiotropy. We did not specifically analyze in how many cases this was not the case, i.e. patterns were consistent with AP in only one replicate. This is now highlighted in the text P5 L20

p. 4, line 18, “No pattern matching...” Please, explain what this means and which results motivates this conclusion. How does this statement square with the legend to Fig. 2, second last sentence, where “conditional (CN) in both replicates” is mentioned?

The wording was changed here to make this statement clear. P5 L30

p. 5 line 10, “markers with breeding values for “number of open flowers”” Please, explain what this means.

Done P6 L20

p. 5, line 31, “either herbivory or pollination for many traits (Table S4)”. Why not include the factors herbivory and pollination in the analysis presented in Table S4, rather than the factor “Biotic treatment”? This would allow the authors to separate the effects of the two treatments.

This is now in Table S5. We have combined pollination and herbivory to the factor “biotic treatments”, because our analysis showed that the two factors together cause the strongest local adaptation. This is also the way the data is presented in Figure 2, where the biotic treatment groups are split.

p. 3, line 3. Wouldn't a priori the strongest evidence of local adaptation had been expected in the “NHH” treatment, where conditions during experimental evolution were likely to be the most similar to those of the “reciprocal transplant experiment” (cf. Fig. 2)? The authors argue that biotic interactions may reinforce trade-offs caused by adaptation to different soil types. At the same time, the results apparently suggest that adaptation to the biotic agents is not associated with a fitness cost in an environment where they are not present.

We agree this is an interesting point, especially for herbivory. For herbivory, one may expect that defense mechanisms that have evolved under herbivory come at a cost and thus lead to decreased fitness in herbivore-free environments. We think the answer for this is in the fact that in our local adaptation experiment, the two environments that plants were transplanted to/from, were identical in terms of biotic interactions (i.e. there were no herbivores present in either limestone or in tuff soil). As our analyses are based on a comparative estimate of fitness in local and foreign environment, the effects of adaptation to the biotic interactors are not seen. We merely see their indirect effect on adaptation to soil type. We agree that studying adaptation to the biotic environment would be a very nice follow up experiment.

p. 6, line 35, “traits showing a pattern of local adaptation”. Unclear what this means. A significant origin x environment interaction (“soil x soil-line interaction” in this case) for an arbitrary trait does not necessarily indicate local adaptation. For such an interaction to indicate local adaptation would require that in each environment, the “local genotype” is characterized by a phenotype closer to the optimal value in that environment. In other words, data on phenotypes are not sufficient; in addition, something needs to be known about the optimal phenotype in the contrasting environments.

We now deleted Table 2

Methods

p. 7, line 22. Were plants grown in individual pots, or in experimental arrays with 49 plants in each? Were plants of a given treatment kept together throughout the experiment, or were they fully randomized except during bee-pollination and herbivory treatments?

We added explanations in this regard in P10 L8

p. 7, line 25, “Plants grown from the seeds of these different crosses...” Unclear what the message of this sentence is. Can be read as if there were no phenotypic differences among treatments, which apparently there were according to the results.

Corrected p9 L25

p. 8, line 13, “to provide overall insights”. Unclear what this means.

“overall” changed to “general”

p. 8, line 15, “and support concepts of how different ecological parameters interact to drive plant evolution”. The authors may want to specify which concepts they have in mind.

“support concepts” was deleted

p. 9, line 4. Pollination was allowed on a single day (10 days after herbivory). At this time not all plants were flowering, and among those flowering, flower number varied. This should result in strong selection on flowering phenology, and more specifically on high flower production early in the season. Are there any data indicating that early flowering start is correlated with high total flower and fruit production in *Brassica rapa* when grown on these substrates also when the growing season is not artificially short? Previous work on the rapid-cycling population has indicated that late flowering start may be associated with high total reproductive success (Agren and Schemske. 1993. *Am Nat*), and also that optimal flowering time will depend on environmental conditions (Johnson, Hamman, Franks. 2022. *AJB*).

See comment above

p. 9 line 28. Explain abbreviations used for the different treatments.

Corrected. P11 L31

p. 9, line 22. Please, clarify procedure. Was the pollen donor randomly chosen among the 28 plants assigned as pollen recipients in the hand-pollination treatment, or was it randomly chosen among all 49 plants in the replicate?

Corrected.

p. 10, line 2. What was the design of the hand-pollination treatment in generation 9 and 10? Were all plants and flowers pollinated by hand this time?

Added. P12 L20

p. 10, line 6. How were seed families selected for inclusion in the reciprocal transplant experiment (40 of 49 families included)? Did in this case all families in a replicate have an equal chance to be included (unlike during the experimental evolution)? If not, the change in pollination treatment in some treatments in generation 9 might influence composition of generation 10.

P12 L24 Seed families were randomly selected.

p. 10, line 15, “*Brassica rapa* is an outcrossing species and therefore its reproductive success is highly correlated to pollinator visitation...” Note that such a correlation does not necessarily reflect variation in mating success, as might be subsumed based on the mentioning of pollinator visitation. It could arise because pollinator visitation is negatively correlated with pollen limitation of seed production. However, it could also arise because large plants with many flowers are more attractive to bumblebees without the need of any variation in pollen limitation.

Thanks for the comment

p. 10, line 31 and onwards. The description of the bumblebee choice experiments is tough reading. Would a graph perhaps help explain the various set-ups?

This text part has been improved for clarity P13

I cannot see that this experiment and its analysis (p. 12) take into account the non-independence of plants that are the product a given replicate of the treatments in the experimental evolution. Will this not result inflated degrees of freedom for the test of treatment effect? The study is set up to examine whether conditions during experimental evolution affects evolutionary outcome, and whether this outcome is consistent across replicates. Shouldn't replicates rather than individual plants be considered independent observations of the effects of different selection environments?

See comment above

p. 10, line 33. Does this describe phenotypic differences observed between treatments during the experimental evolution, as well as differences observed between treatments when grown under common conditions in generation 10?

This part has been changed for clarity.

p. 10, line 37, “each replicate”. Does this refer to replicates in the experimental evolution, or to replicates of the bee-choice assay?

Corrected. This refers to each replicate within treatments and not to replicates of the bee-choice assays. P13

p. 11, line 11. The authors should motivate why these particular traits were chosen for study. Did the authors have any specific predictions regarding evolutionary response to the different selection regimes? In particular, how were these traits expected to influence performance on the two different soil types?

Statement was added P13 L30

p. 11, line 46. Does this describe the analysis of traits scored in generation 10, or something in addition to that? Please, clarify.

Now explained, P14 L25

p. 12, line 2, “relative seed set” as dependent variable. Confusing. From the description above, one gets the impression that seed set was not scored, but only bumble first choices. Please, clarify.

For this, data from other generations were used; now explained P14 L25

p. 12, line 3, “all measured traits”. Please, specify which they were. Twelve traits are listed in Tables S3 and S4. To examine the independent effect of all of these traits on fitness measures in replicates that consisted of 49 plants does not seem meaningful.

For the analysis described here relative seed set was measured in generations 1, 2, and 7, which we used to run selection analyses. Although relative seed set was not available in generation 10, we used bee first choices as a proxy of the relative seed set. Bee first choices is highly correlated to plant reproductive success, which also make this variable relevant for selection analysis. Result of models using relative seed set or first bee choices showed similar results, with flower number being under selection in both models. This result means that bumblebees visited more frequently plants with the most flowers, and that plants with the most flowers were also those producing the most seeds.

p. 12, line 6. A slightly different correlation coefficient is reported on p. 10, line 15.

Corrected

p. 12, line 24. Why were “all traits” examined in an analysis of local adaptation? Only variance in fitness estimates will provide insight into possible local adaptation.

This is true but the traits and their pattern in terms of evolutionary change is still interesting, partly because they are related to fitness or because they show phenotypic differentiation of the plants.

p. 12, lines 30-33. The authors may want to reword this to indicate that a significant interaction with soil does not by itself indicate “local adaptation to soil”. Such an interaction would need to be associated with a higher relative fitness of the “home” genotype than of the “away” genotype on both soils, which is tested in the contrasts described later.

Added “potentially” P15 L16

Table S3, table head, “Plants that evolved in the different soil types were analysed separately, because the analysis contained only plants growing in their native soil type...” This statement is difficult to understand, and should be rephrased. I think it is the word “because” that makes the statement confusing.

changed “analysis” to “dataset”

Reviewer #2 (Remarks to the Author):

The authors address an important and complex question regarding the interaction of abiotic and biotic factors as drivers of plant adaptive evolution. Using a manipulation experiment run over multiple generations with a factorial design for soil, herbivores and pollination, they found that traits of *B. rapa* evolved in response to different selection regimes, and that biotic interactions promoted adaptation to soil. They further dissect the genetic basis of this adaptation process by analysing temporal allele frequency shifts and associating candidate SNPs to genetic trade-offs. The study relies on an impressive set of experiments conducted over multiple generations, which, coupled with trait measurements and sequencing, make for a great dataset. Findings are potentially highly interesting and unique in dissecting the role of interactions contributing to evolution, a perspective that is rarely seen in the field. However, the system is complex and results are numerous, which calls for a clear text to allow the reader to follow. The text is currently written in a short format (I guess shorter than the requirements from Nature Communications) but in my opinion, more information and clarity are needed to make the content accessible. Some of the key conceptual statements that underly the study should also be revised. The manuscript could present a highly relevant contribution following revision.

The manuscript aims to address the role of biotic and abiotic factors in driving adaptation, but which evolutionary response is the focus of the study is unclear. In the introduction, the problem is pitched on the role of soil (abiotic) in driving the response to selection imposed by herbivores and pollinators (biotic; P2L1-7). However, later the main question seems to be how biotic factors impact adaptation to soil (P2L11-12). The repeated switch between these two perspectives in the manuscript makes it difficult to follow. I expand on these perspectives below.

The experimental evolution analyses (Fig1, TableS3) look at how selection imposed by biotic factors drives phenotypic shifts, and while experiments are run on separate soils and there is no test of soil x biotic interactions, the text keeps emphasising how the evolutionary responses vary (without statistical evidence) across soils. I do not understand while soil interaction is not tested, as this seems to be a crucial point, not just because the text hints at it repeatedly and the way Fig1 is structured (showing contrasts across soils), but because it is at the basis of the factorial design that is emphasised in the methods. In this regards, I do not understand the statement at P2L31-33. One could guess that many soil interactions are significant, such as the example reported for opposite pollination effects on “Time to first flower” (P2L25; Table S3). Overall, my take home message of this experiment is that soil mediates evolutionary responses to biotic factors (though again, there is no formal test for this, but this is what I deduce from the text).

We have added now the analysis showing this, i.e. interactions between soil and herbivory and soil and pollination (Table S3)

Next, in the transplant experiment (Fig2), the agent of selection switches from biotic factors to soil, and the focus is now on soil adaptation as mediated by biotic factors. Indeed, this fits what is stated as the main goal at P2L10-122. Results here are interesting, but their interpretation in the discussion is only quickly touched upon. Most of the arguments related to this experiment are instead in line with the

perspective of the first experiment (e.g. P5L32 [but not tested], P5L42 and 43; P6L5-8). Overall, I think the misalignment of hypotheses, experiments, and interpretations across the manuscript leaves the reader wondering what question is actually addressed. I realise that dissecting the role of these interaction is complex, but the way this is implemented in the analyses and in the text does not seem coherent, and the text could be improved to sharpen the focus.

This has now been clarified in the introduction and discussion.

Genomic analyses of allele frequency shifts and genetic trade-offs are very interesting but I think more explanations of the results are necessary. The manhattan plots show significant outliers across the whole genome, which calls for a plausible explanation to rule out the suspect of a large number of false positives. Some regions seem to have a clearer signal than others, as identified by discrete peaks emerging from what seems to be a noisy background signature. I understand that initial commercial lines were genetically diverse and plants are self-incompatible, so I imagine LD is low in this system? I guess SNP signal can be very noisy and a window analysis may yield more discrete evidence. You could still take SNPs within significant windows for the AP vs CN downstream tests. Alternatively, a neutral expectation through simulations may be needed here.

We have changed the SNP analysis now to a window-based approach. We chose 2Kb windows according to a linkage decay analysis ². This reduces the number of significant SNPs, but did not change the number and identity of genes associated to SNPs with a pattern of antagonistic pleiotropy, showing that the analysis is robust to this approach of reducing the number of (linked) SNPs.

The second part of the genomic analyses relies on the BLUP estimate of SNP fitness effects. More information is needed in the methods to assess how the analyses were run and significance of the results. Why was a training set used, isn't the number of flowers available for all individuals? I can't understand how the sample size for each treatment and replicate can be > 49? Does it mean that predictions were run on 45-72 samples independently 16 times, and how were these data combined to give values plotted in Fig3? This seems a very low sample size for such predictions. Effects can certainly have high values but also very large SE. Was any filtering for significance applied? A table with effect, SE and significance should be provided. It would also be very interesting to see these effects combined with the manhattan plot. This could address one of your statements in the introduction (P1L36) of unclear links between genetic trade-offs and divergence. I guess the expectation would be that strong divergence should emerge from AP effects?

The training set is a part of the machine learning analysis used in the BLUP package. The sample size greater than 49 was achieved by analyzing plants growing in local and foreign soil together; thus the maximum sample size was 72. This now mentioned in the text (P16 L17). We chose not to provide any information on effect sizes of loci, as the breeding values were merely used to filter for SNPs which have an association to the trait "number of flowers". No analysis of effect sizes was included in the paper, as this was considered not within the scope in this study.

Figure 1

As mentioned above, Fig1 is misleading and I find it difficult to match to the text. In comparison, Table S3 is easier to look at. Perhaps you could consider combining plots and statistics? I think plots would be clearer separated by soil (instead of combined). What are the bars on top of each column?

We have adapted and modified Figure 1 now accordingly; bars are s.e.m. -now mentioned on the axis.

Sum of visits is not reported in TableS3. Bars are missing for these columns.

These are total sums; thus there is no variation. The values are not in Table S3 as the statistics differ because of the experimental setup. Instead, we show a figure of the data (Figure S3).

As far as I can figure use of letters for significance relies on fixed effects of TableS3, so their use is different from what is commonly seen, which is individual contrasts of all values represented in the plot. For example, for Number of flowers in Tuff, BB seems to indicate that two aphid treatments are not significant, while they are significant from columns indicated with A. I also don't understand why G1 receives a letter and results from this contrast are not presented in the text. Please clarify.

This has now been fixed; letters were replaced by asterisks in Figure 1. Statistical supports for the figures can be found in in Table S3 for bee choices and Table S4 for flower number and flower diameter.

Figure 2 How are manhattan plots generated? There are 2 replicates analysed independently, so what is represented here?

The qqman package was used for the Manhattan plots (now mentioned P16 L2). Replicates are analyzed together here, as only SNPs with significant values in both replicates were included.

P1L36-40: many concepts in 4 lines without explanation. Some of these do not seem relevant for this study as they are not further addressed (e.g. ecological speciation).

This has been re-written; speciation is discussed in the discussion now.

P1L32+38: clarify these statements relevant for the manuscript, we do know something about these processes

More on this topic has been added to the manuscript

P1L36: the link between antagonistic selection and AP vs CN is not clear

This text part has been modified

P1L46: report what is known about adaptation to pollinators and herbivores, or on the clarified study questions

Done, P2 L16

P2L20: 'native' soil is not defined at this point as it comes only in the methods

This text part was re-written

P2L11-20: introduction of traits or fitness measured is needed to be able to follow results

Done

P2L21: what's the evidence for resource reallocation?

Here, the evidence is some traits increased whereas others increased

P2L24-40: descriptions difficult to follow

The part was re-written

P3L15: reciprocal transplant experiment has not been presented before

This is now introduced P3 L8

P3L36: results difficult to follow

Now re-written

P4L4: pvalue missing

Replaced by "test"

P4L19: what does this mean: "pattern that matches phenotypic data"?

Wording changed P5 L28

P5L32: this is not tested

Tests in this regard were added in Table S3. Furthermore, we used Table S4 to simplify the interpretation of Table S3. Although, we included soil line in this table, it is important to point out that plants were grown in different soils, and we can therefore not distinguish “soil genotype effect” from soil plasticity, complicating the interpretation solely based on the “soil line” factor.

P12L38-44: home vs away is not reported in results

Home versus away is indicated for number of open flowers in Table 1. For other traits, results for both contrasts (home vs away and foreign vs local) are presented in Table S5 to S8.

- 1 Ashman, T. L., Galloway, L. F. & Stanton, M. L. APPARENT VS EFFECTIVE MATING IN AN EXPERIMENTAL POPULATION OF RAPHANUS-SATIVUS. *Oecologia* **96**, 102-107, doi:10.1007/bf00318036 (1993).
- 2 Frachon, L. & Schiestl, F. P. Rapid genomic evolution in *Brassica rapa* with bumblebee selection in experimental evolution. *Bmc Ecology and Evolution* **24**, doi:10.1186/s12862-023-02194-y (2024).

REVIEWERS' COMMENTS

Reviewer #1 (Remarks to the Author):

I think this the manuscript has become greatly improved thanks to the revisions made by the authors. Several aspects of experimental procedures and interpretations have been clarified, which makes the presentation much more accessible. I think it is an interesting paper, and I have only a few comments at this point.

p. 3, line 122, "We expected from earlier studies ... that pollinator-mediated selection should vary according to soil type..." The authors should indicate how pollinator-mediated selection was expected to vary according to soil type. Without this information, the logic behind the conclusion formulated in the following sentence ("Therefore, ...") is unclear.

Fig. 1 legend, line 159, "significant effects of soil". Unclear which asterisks in the graph indicate significant effects of soil. Are there any?

Fig. 1 legend, line 161, "... was tested separately for soil- and herbivory groups". This is difficult to understand. Does it mean that the effect of pollination treatment was tested separately for each combination of soil and herbivory treatment?

p. 4, line 174, "...for "time to first flower", the performance pattern matched the local vs. foreign criterion". Unclear what this means. Do the authors assume that early flowering is associated with high fitness in both soil environments?

p. 5, line 199, "...there was significant overlap between replicates of the two treatment groups, as only markers that showed an AP-pattern in both replicates were considered in the analysis". Logic a bit unclear. Wouldn't it be interesting to consider also those markers that showed an AP-pattern in only one replicate to be able to judge which category was more common (AP pattern observed in one vs. both replicates of each treatment)?

p. 5, line 210, "... we assumed that local adaptation was driven by AP rather than CN." Total contribution of AP and CN to local adaptation will depend on number of AP and CN loci, respectively, but also on effect sizes (and for CN, direction of effect on fitness). Shouldn't therefore also effect sizes and directions be considered in assessment of the relative importance of AP and CN for the overall pattern of local adaptation?

Fig. 5. It is difficult to distinguish the colour codes used for "Tuff line" and "Limestone line", and also those that should distinguish "Neither CN nor AP" and "CN".

Fig. 5. Another detail: Why not reorder symbols such that graphical symbols and letters indicating pollination and herbivory treatments are listed in the same order in both cases?

p. 6, line 256, "A recent study...". Is that paper reporting results from the same experiment as presented here? I think this should be stated explicitly.

p. 10, line 441. Shouldn't variation among individuals in contribution to the next generation be considered when estimating effective population sizes?

p. 11, line 473. Why this shift in how pairs were formed for hand-pollination compared to earlier generations?

p. 11, line 487. This correlation is slightly different from the one reported on line 578. The correlation based on data in the present study would seem to be the most relevant to report?

p. 12, line 530. Unclear what is meant by "a complementary approach". Complementary to what?

p. 12, line 539, "Other traits...". This sentence was difficult to follow. Could potentially be fixed by a

few changes:

line 541. Insert "start of" before "flowering"

line 541. Change "plants" to "plant".

line 541. Insert "length of" before "branches"

line 542. Delete "measurements", and change "done" to "recorded".

p. 12, line 542. Was a single leaf measured on each plant? If so, which leaf was chosen for measurement?

p. 13, line 585. Isn't it more common to use total flower production rather than "number of open flowers" at a given time as a fitness proxy?

p. 13, line 593, "does not correlate with". Change to "is not strongly correlated with"?

p. 15, line 652, "... significant value of AF change for ... or in only one soil type was discarded". Doesn't this remove loci potentially contributing to conditional neutrality?

p. 15, line 675. For how many of the loci showing a signature CN was the predicted effect on number of open flowers in the direction that the change in allele frequency would contribute to local adaptation?

p. 1, line 31, "of". Change to "with"?

Reviewer #2 (Remarks to the Author):

I thank the authors for addressing my points on the soil interactions and the type of selection investigated, I believe the manuscript has greatly improved. I find this manuscript revealing of the complexity of processes underlying local adaptation, specifically to soil, but also in a broader perspective. It is certainly interesting that biotic components facilitate soil adaptation, which is well emphasised in the discussion, but it's equally interesting that soil alone does not seem to impose sufficient selection to drive adaptation (although it may just take longer). I feel this aspect leaves open questions for the reader, carrying important implications for the many studies addressing soil adaptation. It's however now understated in the discussion, and I suggest the authors take the opportunity to share their view on this. Can soil alone be an agent of selection? What do we know from other systems? What's the broader implication of this study, should we hypothesise that soil adaptation is inherently mediated by biotic factors?

Because of the potential impact of this study, I think all its parts should read convincing. The BLUP analyses use a very specific methodology, and they are crucial for the AP vs CN results. My question was whether the estimated breeding values have a confidence interval or a pvalue that prove them significant, and thus meaningful to retain SNPs. I believe the authors should provide the reader with better information to understand these analyses.

The introduction is now clear on the concepts but I feel that it misses the genomic part related to the AP vs CN concepts laid out at the start. The aims of the study would be clearer if presented with explicit questions.

P4L173 As stated at L177, patterns on traits can't be referred to local adaptation, so it's confusing to apply the terminology of local vs foreign etc.

P7L300 This sentence does not work. Please rephrase.

RESPONSE TO REVIEWERS' COMMENTS

Reviewer #1 (Remarks to the Author):

I think this the manuscript has become greatly improved thanks to the revisions made by the authors. Several aspects of experimental procedures and interpretations have been clarified, which makes the presentation much more accessible. I think it is an interesting paper, and I have only a few comments at this point.

p. 3, line 122, "We expected from earlier studies ... that pollinator-mediated selection should vary according to soil type..." The authors should indicate how pollinator-mediated selection was expected to vary according to soil type. Without this information, the logic behind the conclusion formulated in the following sentence ("Therefore, ...") is unclear.

Sentences were reformulated to clarify the logic behind our hypotheses. P3 L129

Fig. 1 legend, line 159, "significant effects of soil". Unclear which asterisks in the graph indicate significant effects of soil. Are there any?

Significant effects of soils were added. P20 L938

Fig. 1 legend, line 161, "... was tested separately for soil- and herbivory groups". This is difficult to understand. Does it mean that the effect of pollination treatment was tested separately for each combination of soil and herbivory treatment?

Here we referred to soil types and not soil evolutionary lines. We now indicate "soil types" to avoid confusion. P10 L941

p. 4, line 174, "...for "time to first flower", the performance pattern matched the local vs. foreign criterion". Unclear what this means. Do the authors assume that early flowering is associated with high fitness in both soil environments?

We deleted „performance“ to make clear we mean only the pattern and make no inferences to fitness effects P4 L173

p. 5, line 199, "...there was significant overlap between replicates of the two treatment groups, as only markers that showed an AP-pattern in both replicates were considered in the analysis". Logic a bit unclear. Wouldn't it be interesting to consider also those markers that showed an AP-pattern in only one replicate to be able to judge which category was more common (AP pattern observed in one vs. both replicates of each treatment)?

Yes, but as we used the CMH test for assessing significant changes, only markers that showed similar AF changes in both replicates are included, and thus the information of marker behavior in individual replicates is lost. More detailed analysis of AP were not desired for this study.

p. 5, line 210, "... we assumed that local adaptation was driven by AP rather than CN." Total contribution of AP and CN to local adaptation will depend on number of AP and CN loci, respectively, but also on effect sizes (and for CN, direction of effect on fitness). Shouldn't therefore also effect sizes and directions be considered in assessment of the relative importance of AP and CN for the overall pattern of local adaptation?

Information on effect sizes were now added in the last paragraph of page 5 (P5 L218).

Fig. 5. It is difficult to distinguish the colour codes used for "Tuff line" and "Limestone line", and also those that should distinguish "Neither CN nor AP" and "CN".

The color logic (blue limestone and green for tuff) follows the rest of the MS; we hope that the print setter will ensure these colors can be readily distinguished (P21 L950).

Fig. 5. Another detail: Why not reorder symbols such that graphical symbols and letters indicating pollination and herbivory treatments are listed in the same order in both cases?

Thank you for the suggestion, we reordered the symbols to match treatments coding (P21 L950).

p. 6, line 256, "A recent study...". Is that paper reporting results from the same experiment as presented here? I think this should be stated explicitly.

This sentence refers to a different experiment. We have added "another" to makes this clear (P6 L264).

p. 10, line 441. Shouldn't variation among individuals in contribution to the next generation be considered when estimating effective population sizes?

Technically yes, but this variation was small and an overall value was needed for the statistical assessment of significant AF change between generation one and ten.

p. 11, line 473. Why this shift in how pairs were formed for hand-pollination compared to earlier generations?

In previous generation, 28 plants were randomly selected for hand-pollination. In generation 9, we decided to hand-pollinated all plants, to ensure that genetic variation was preserved across both the hand-pollinated and bee-pollinated groups.

p. 11, line 487. This correlation is slightly different from the one reported on line 578. The correlation based on data in the present study would seem to be the most relevant to report?

Corrected (P12 L571).

p. 12, line 530. Unclear what is meant by "a complementary approach". Complementary to what?

Complementary changed to „comprehensive“ (P10 L476).

p. 12, line 539, "Other traits...". This sentence was difficult to follow. Could potentially be fixed by a few changes:

line 541. Insert "start of" before "flowering" done (P11 L531).

line 541. Change "plants" to "plant". done (P12 L531).

line 541. Insert "length of" before "branches" done (P12 L532).

line 542. Delete “measurements”, and change “done” to “recorded”. done (P12 L533).

p. 12, line 542. Was a single leaf measured on each plant? If so, which leaf was chosen for measurement?

We measured only the width and length of the first true leaf. Details added (P11 L530).

p. 13, line 585. Isn't it more common to use total flower production rather than “number of open flowers” at a given time as a fitness proxy?

Because pollination only occurred on a given date for all plants, pollinators only interacted with the number of open flowers at this particular date. This fact led our decision in considering flower number rather than flower production. Furthermore, our results across generations provided support for this choice as number of flowers could predict number of pollinators visits and seed set (Supplementary Table 1).

p. 13, line 593, “does not correlate with”. Change to “is not strongly correlated with”?

This was changed to “not significantly correlated to” (P13 L587).

p. 15, line 652, “... significant value of AF change for ... or in only one soil type was discarded”. Doesn't this remove loci potentially contributing to conditional neutrality?

This was done to ensure only markers with consistent AF change values were used in the analysis. A pattern of markers only significant in one replicate is more likely to be caused by drift. Markers contributing to CN were only considered if they did so in both replicates.

p. 15, line 675. For how many of the loci showing a signature CN was the predicted effect on number of open flowers in the direction that the change in allele frequency would contribute to local adaptation?

Following this comment, we report now the effect sizes for markers with AP and CN pattern in the text (last paragraph of results section: P5 L216).

p. 1, line 31, “of”. Change to “with”?

done (P1 L31).

Reviewer #2 (Remarks to the Author):

I thank the authors for addressing my points on the soil interactions and the type of selection investigated, I believe the manuscript has greatly improved. I find this manuscript revealing of the complexity of processes underlying local adaptation, specifically to soil, but also in a broader perspective. It is certainly interesting that biotic components facilitate soil adaptation, which is well emphasised in the discussion, but it's equally interesting that soil alone does not seem to impose sufficient selection to drive adaptation (although it may just take longer). I feel this aspect leaves open questions for the reader, carrying important implications for the many studies addressing soil adaptation. It's however now understated in the discussion, and I suggest the authors take the opportunity to share their view on this. Can soil alone be an agent of selection? What do we know

from other systems? What's the broader implication of this study, should we hypothesise that soil adaptation is inherently mediated by biotic factors?

We have added a paragraph in the discussion addressing this interesting point (2nd last paragraph, discussion: P7 L289)

Because of the potential impact of this study, I think all its parts should read convincing. The BLUP analyses use a very specific methodology, and they are crucial for the AP vs CN results. My question was whether the estimated breeding values have a confidence interval or a pvalue that prove them significant, and thus meaningful to retain SNPs. I believe the authors should provide the reader with better information to understand these analyses.

The rrBLUP does not produce a CI of P-values for breeding values.

The introduction is now clear on the concepts but I feel that it misses the genomic part related to the AP vs CN concepts laid out at the start. The aims of the study would be clearer if presented with explicit questions.

Corrected P3 L90

P4L173 As stated at L177, patterns on traits can't be referred to local adaptation, so it's confusing to apply the terminology of local vs foreign etc.

The terminology local versus foreign is essentially comparative in this case where we compare the values in local vs. foreign plants. Although, this terminology is commonly associated with local adaptation, we believe it accurately reflects our comparative approach. However, as stated by reviewer 1, our use of "performance pattern" is misleading / confusing, as performance can be expressed as fitness-related traits. Therefore, we erased performance of this sentence to avoid any ambiguity (P4 L173).

P7L300 This sentence does not work. Please rephrase.

done (P7 L329).